DOI: 10.1038/s41467-018-03597-y | **OPEN**

# Dynamical origins of heat capacity changes in enzyme-catalysed reactions

Marc W. van der Kamp [1,2], Erica J. Prentice [3], Kirsty L. Kraakman[3], Michael Connolly [2], Adrian J. Mulholland[2] & Vickery L. Arcus [3]

Heat capacity changes are emerging as essential for explaining the temperature dependence of enzyme-catalysed reaction rates. This has important implications for enzyme kinetics, thermoadaptation and evolution, but the physical basis of these heat capacity changes is unknown. Here we show by a combination of experiment and simulation, for two quite distinct enzymes (dimeric ketosteroid isomerase and monomeric alpha-glucosidase), that the activation heat capacity change for the catalysed reaction can be predicted through atomistic molecular dynamics simulations. The simulations reveal subtle and surprising underlying dynamical changes: tightening of loops around the active site is observed, along with changes in energetic fluctuations across the whole enzyme including important contributions from oligomeric neighbours and domains distal to the active site. This has general implications for understanding enzyme catalysis and demonstrating a direct connection between functionally important microscopic dynamics and macroscopically measurable quantities.

[1] School of Biochemistry, Biomedical Sciences Building, University of Bristol, University Walk, Bristol BS8 1TD, UK. [2] Centre of Computational Chemistry, School of Chemistry, University of Bristol, Cantock's Close, Bristol BS8 1TS, UK. [3] School of Science, University of Waikato, Hamilton 3216, New Zealand. These authors contributed equally: Marc W. van der Kamp, Erica J. Prentice. Correspondence and requests for materials should be addressed to M.Kamp. (email: marc.vanderkamp@bristol.ac.uk) or to A.J.M. (email: adrian.mulholland@bristol.ac.uk) or to V.L.A. (email: varcus@waikato.ac.nz)

A critical variable for the rate of a reaction is temperature. For uncatalysed chemical reactions, the rate of reaction typically increases exponentially with increasing temperature, as described by the Arrhenius and Eyring equations[1,2]. In reactions catalysed by enzymes, the effects of temperature are complex and include (often opposing) contributions from active site geometry and reactivity, protein stability, conformational changes and temperature-dependent regulation. Changes in temperature can also potentially affect features of the enzyme-catalysed reaction outside the chemical steps, such as substrate binding, product release and conformational changes. Despite these complexities, enzymes generally show a characteristic temperature profile including an optimum temperature ($T_{opt}$) for activity above which rates decline with increasing temperature. The decline in rate above $T_{opt}$ cannot simply be explained by enzyme unfolding at higher temperatures and deviations from Eyring behaviour are also often seen at temperatures below $T_{opt}$[3–5]. We recently developed macromolecular rate theory (MMRT)[6,7], which explains the temperature dependence of enzymes including an intrinsic $T_{opt}$ in the absence of denaturation by introducing the concept of heat capacity changes along the reaction coordinate: the heat capacity ($C_P$) for the enzyme–substrate complex is generally larger than $C_P$ for the enzyme–transition state (TS) complex, in enzymes for which the chemical reaction is rate limiting. Hence, the activation heat capacity, $\Delta C_P^{\ddagger}$, for the enzyme-catalysed reaction is generally negative (Fig. 1a; in case the chemical reaction is not rate limiting, a small positive $\Delta C_P^{\ddagger}$ is possible[8]). We have demonstrated that this accounts for the curvature observed in Eyring plots for a number of enzymes[6]. As we have discussed previously, curvature in Eyring plots due to $\Delta C_P^{\ddagger}$ has also been observed for protein-folding kinetics (e.g. ref. [9]). This is directly analogous to temperature-dependent curvature in protein stability due to $\Delta C_P$ that gives rise to both high- and low-temperature denaturation[10].

$\Delta C_P^{\ddagger}$ is a statistical thermodynamic property for the catalysed reaction that describes the difference in heat capacity between the thermodynamic ensemble in the ground state and that at the TS. It can be determined experimentally[11], and can also be calculated from the variance in enthalpy at equilibrium for each of these states[12]

$$\Delta C_P^{\ddagger} = \frac{\Delta \langle \partial H^2 \rangle^{\ddagger}}{k_B T^2}. \tag{1}$$

In principle, atomistic molecular dynamics (MD) simulations at equilibrium can provide a distribution of enthalpies from which the variance $\langle \partial H^2 \rangle$ (the mean square fluctuation in the enthalpy) may be calculated. To do so, the ensemble for the enzyme–substrate complex and separately, that for the enzyme–TS complex, should be simulated.

Here we experimentally determine the value for $\Delta C_P^{\ddagger}$ from the temperature dependence of the rate in the absence of enzyme denaturation for two quite different enzymes: the small, dimeric ketosteroid isomerase (KSI) and the large, monomeric α-glucosidase MalL. In parallel, we employ extensive MD simulations (10 μs per enzyme) to obtain heat capacity differences between two states along the reaction pathway. KSI is a very well-studied enzyme that is involved in steroid biosynthesis and degradation: it performs two consecutive proton transfers to shift the position of a C=C double bond[13]. MalL is a large α-glucosidase: it hydrolyses terminal non-reducing (1 → 6)-linked α-glucose residues in a two-step reaction, releasing α-glucose[14]. Previously, we have shown by experiment that there is a large

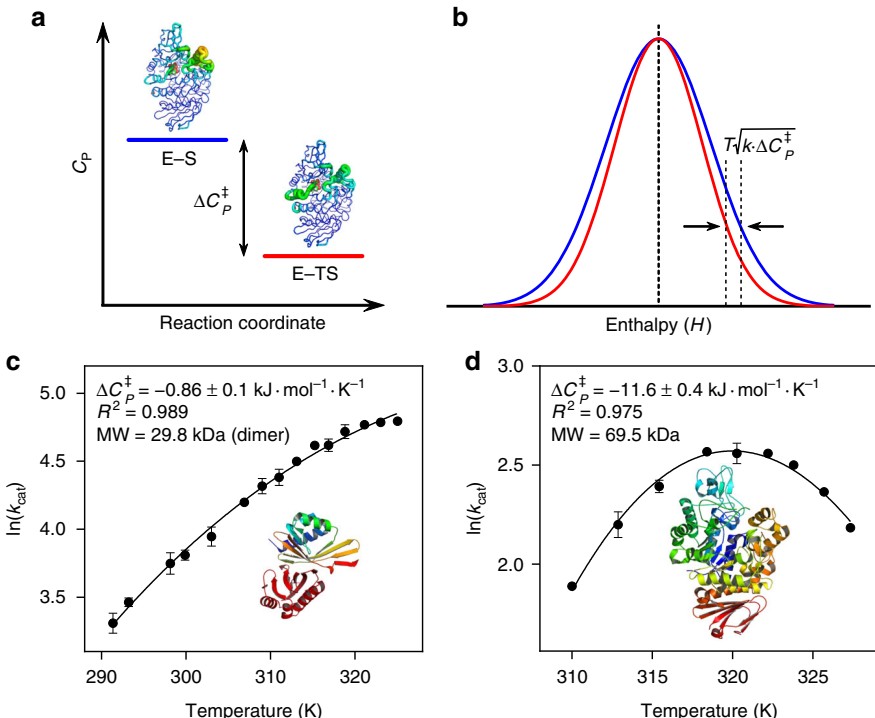

**Fig. 1** Basis of a negative $\Delta C_P^{\ddagger}$ and its determination through experiment or simulation. **a** Conceptual depiction of a difference in $C_P$ between the enzyme–substrate (E–S) and enzyme–transition state (E–TS) complexes along a reaction, resulting in a negative $\Delta C_P^{\ddagger}$. **b** Conceptual depiction of differences in enthalpy distribution at the E–TS (red) and the E–S states (blue). Arrows indicate the inflection points (at $\mu + \sigma$), and the difference defines $\Delta C_P$ between the two states according to the formula given (see Eq. 1). **c-d** Experimentally determined $\Delta C_P^{\ddagger}$ values (kJ mol$^{-1}$K$^{-1}$ ± SE) for the temperature-dependent rates of KSI (**c**) and MalL (**d**). The data are fit with MMRT (see Methods). Error bars, where visible, represent the standard deviation of three replicates. Structures of KSI and MalL are drawn to scale

change in heat capacity for this enzyme-catalysed reaction and that single point mutations can dramatically alter the temperature dependence of the rate by altering the heat capacity of either the enzyme–substrate complex or the enzyme–TS complex[7].

Activation heat capacities from simulations and experiment are in good agreement for both enzymes. This shows that prediction of activation heat capacity for enzymes is feasible by simulations, opening a new route to predicting and engineering optimum temperatures for enzyme activities. Further, the simulations provide an atomically detailed picture of the dynamical differences between TS and Michaelis complexes that give rise to this behaviour, revealing complex, and intriguing changes in dynamics across the whole enzyme structure (similar to changes observed by nuclear magnetic resonance in protein dynamics upon ligand binding[15–17]). We thus use simulation to interpret the $\Delta C_P^{\ddagger}$ obtained from macroscopic kinetics measurements in terms of detailed contributions at the atomistic level, providing a link between enzyme structural and energetic molecular fluctuations to its function and thermoadaptation.

## Results

$\Delta C_P^{\ddagger}$ **determined by experiment:** As shown previously[6,7], $\Delta C_P^{\ddagger}$ can be determined by fitting the ln(rate)-versus-temperature plot using MMRT (Fig. 1). For MalL, curvature in this plot is very significant (Fig. 1d), and unrelated to unfolding[5]. This leads to a negative $\Delta C_P^{\ddagger}$ value of $-11.6 \pm 0.4$ kJ mol$^{-1}$ K$^{-1}$. For KSI, the curvature is less extreme, but still obvious, leading to a small negative $\Delta C_P^{\ddagger}$ of $-0.86 \pm 0.1$ kJ mol$^{-1}$ K$^{-1}$ (Fig. 1c). An important consequence of the $\Delta C_P^{\ddagger}$ values for each enzyme is the position of the optimum temperature ($T_{opt}$) for activity as these parameters are correlated. For example, the large negative $\Delta C_P^{\ddagger}$ value for MalL dictates the position of $T_{opt}$ at 320 K, whereas the much smaller $\Delta C_P^{\ddagger}$ value for KSI places the $T_{opt}$ well above 320 K (in absence of protein unfolding). Similarly, the significant curvature of the MalL temperature dependence means that at lower temperatures, the rate approaches zero much faster for MalL than for KSI. Implicit in this approach is the assumption that $\Delta C_P^{\ddagger}$ is independent of temperature in the temperature range studied, and is justified based on the good fit of the MMRT model to the experimental data.

**Heat capacity differences from simulation.** Heat capacity differences for enzyme-catalysed reactions can be calculated from $\Delta \langle \partial H^2 \rangle^{\ddagger}$ (Eq. (1)). To measure $\Delta \langle \partial H^2 \rangle^{\ddagger}$ from simulation, there are two main challenges: (a) the amount of sampling required for the system to define the enthalpy variance, and (b) an accurate and consistent representation of the reactant state (Michaelis complex) and the TS. A statistical thermodynamic analysis of a 1 ms MD simulation of the bovine pancreatic trypsin inhibitor indicated that 10s of microseconds (μs) of simulation may be needed to converge the heat capacity difference between two conformational states[18]. Sampling on the order of (at least) microseconds is thus expected to be required for reliable identification of heat capacity differences. Such sampling is now routinely feasible with a 'molecular mechanics' description of the atoms and their interactions. Molecular mechanics force fields have been developed and optimised over many years[19], and can provide a generally good description of the structure and dynamics of proteins and protein–ligand binding[20]. They are, however, empirical potential functions and typically (for reasons of computational efficiency) lack physically important effects, such as variations in electronic polarisation. This may be related to limitations in their ability to capture details of fast dynamics[21]. Here, to calculate $\Delta C_P^{\ddagger}$, we use extensive MD simulation to quantify the change in fluctuation between two states, A and B (Fig. 1a). The difference in heat capacity between these states can be determined by:

$$\Delta C_{A,B} = \frac{\langle \delta H_B^2 \rangle - \langle \delta H_A^2 \rangle}{k_B T^2}. \tag{2}$$

To sample the conformational dynamics of the reactant (E–S or RS) and 'TS' (E–TS) enzyme complexes consistently, electronically unstable states (e.g. with half-formed bonds involving enzyme residues) should be avoided for the 'TS' representation. We, thus, use molecular species that are representative for the TS (i.e. TS analogues), and predict that these will show a similar heat capacity change from the reactant state. This prediction has been demonstrated experimentally for human 5′-methylthioadenosine phosphorylase[11]. For KSI, a charged enediolate intermediate is formed after the first proton transfer, and this is the key species stabilised by the enzyme for catalysis of the reaction[22,23]. We use this intermediate state as a proxy for the two enzyme–TS complexes (one for each proton transfer) as the intermediate lies between the two TSs at similar energy. The substrate (5-androstene-3,7-dione) and intermediate complexes (Fig. 2a, Supplementary Fig. 1; Supplementary Fig. 3) were built based on KSI in complex with the inhibitor 5α-estran-3,17-dione (PDB 1OHP). For MalL, we obtained an experimental X-ray structure co-crystallised with a stable transition-state analogue (Supplementary Table 1; Supplementary Fig. 4 and 5) and use this to simulate the thermodynamics of the substrate isomaltose and a close analogue of the transition-state species at the rate determining step[24] (Fig. 2b; Supplementary Fig. 2).

A total of 5 μs of explicit solvent MD simulation was run for KSI and MalL in both the substrate-bound and proxy TS representations over ten replicate simulations for each state. The force-field potential energy was used as an approximation for the system enthalpy, and was recalculated for the protein–ligand system (i.e. all atoms) without explicit water. Considering the variance of the enthalpy is the quantity required for $C_P$ calculation (Eq. (1)) and a difference in variance between two states is used to determine $\Delta C_P^{\ddagger}$ (Eq. (2)) these approximations should be reasonable. Calculating the variance with explicit solvent is problematic because there is no clear criterion for selecting the water molecules that should be included in such a calculation (see Supplementary Note 6). We note that $\Delta C_P^{\ddagger}$ values calculated with an implicit solvent are qualitatively similar to those reported below for both enzymes (Supplementary Table 5). Note that there is, in principle, an alternative approach to calculating $\Delta C_P^{\ddagger}$, via the variance in entropy[12]. Calculating entropy from simulation accurately is much more challenging; however, this may be feasible in the future.

For KSI, the conformational space sampled is limited, with only two distinct structural clusters discernible (see Supplementary Note 5, Supplementary Fig. 6 and Supplementary Table 3). The difference between these clusters is in a small region in the unoccupied monomer (Fig. 2c). The $H$ variance is significantly different between the clusters, however (Fig. 2e). For $\Delta C_P^{\ddagger}$ calculation, we thus calculate the variance of the clusters separately, with the total variance for each state being the average variance weighted by the cluster occupation (Fig. 2e; Supplementary Note 5).

MalL samples a larger conformational space than KSI, occupying and regularly switching between a number of structural clusters along the simulation trajectories, related primarily to changes in loops surrounding the active site (Fig. 2d; Supplementary Notes 4 and 5, Supplementary Table 2, Supplementary Fig. 8, 10 and 11). Due to the presence of multiple

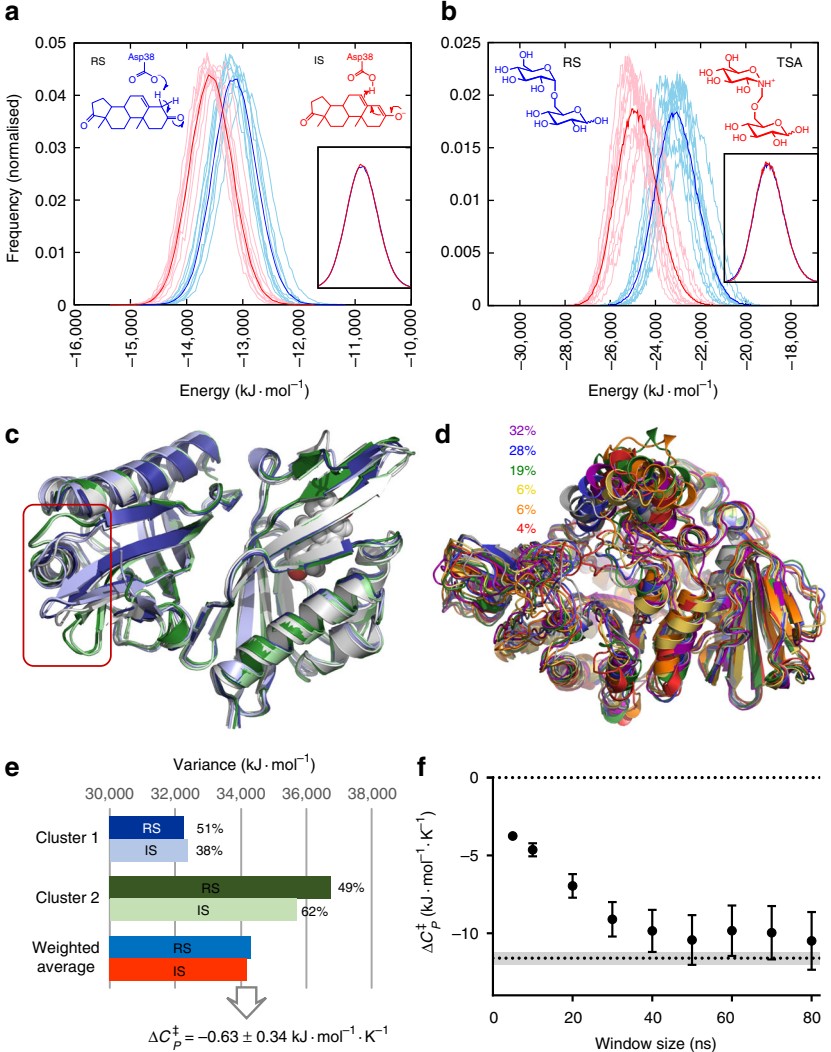

**Fig. 2** Sampling and $\Delta C_P^{\ddagger}$ calculation in simulations. **a-b** Histograms of energies from 50 to 500 ns MD simulations for KSI (**a**) and MalL (**b**). Thin lines are individual runs, thick lines are the average of ten runs. Insets show overlay of histograms for both states, and the structures indicate the species simulated (RS reactant state, IS intermediate state, TSA transition state analogue). **c** Representative structures for the two distinct conformational clusters present in the KSI simulations of both states (reactant state in blue and green, intermediate state in pale blue and green, starting structure in light grey). Box highlights the region with structural differences. **d** Representative structures for the six main conformational clusters in MalL reactant state simulations and their occupancies (starting structure in light grey). **e** Variance in energies for the two clusters identified in the KSI simulations, with cluster occupancies (in %) and weighted average variance for both states. **f** Convergence with moving average-window size of $\Delta C_P^{\ddagger}$ values calculated for MalL, with value determined from experiment indicated by dotted line (with grey area indicating standard deviation). Error bars indicate the standard deviation of the calculated $\Delta C_P^{\ddagger}$ values based on the cumulative standard deviation for each state, from 10 independent simulations

conformational clusters, consideration of the system over the full simulation time overinflates the enthalpy variance. However, calculating variances for each cluster (as for KSI) does not take into account that frequent switches between the distinct conformational states will also contribute to the variance. In addition, several clusters are dominated by one state only (Supplementary Fig. 11). To be independent of clustering and account for switching between conformational substates, enthalpy variance was calculated using a moving window along the simulation trajectory for each simulation, and subsequent averaging. The 'window' for the moving average was varied between 5 and 80 ns and calculated $\Delta C_P^{\ddagger}$ values converge when the window size is between 40 and 80 ns (Fig. 2f, Supplementary Table 4). Thus, the calculated $\Delta C_P^{\ddagger}$ values for MalL converge on a value of $-10.0 \pm 1.7 \text{ kJ mol}^{-1} \text{ K}^{-1}$ (using a window of 70 ns), which is within the error range of the experimentally determined $\Delta C_P^{\ddagger}$ value of $-11.6 \pm 0.4 \text{ kJ mol}^{-1} \text{ K}^{-1}$.

**Local and global contributions to $\Delta C_P^{\ddagger}$.** The observation that $\Delta C_P^{\ddagger}$ values calculated from extensive conformational sampling agree with those determined experimentally allows meaningful analysis of the differences between the two ensembles. A significant part of $\Delta C_P^{\ddagger}$ resides in the protein backbone (in agreement with experiments that suggest side-chains contribute only a fraction to the total protein heat capacity[17]; Supplementary Table 5), although the contribution of side-chains cannot be ignored. Striking results emerge from analysing contributions from different regions of the enzymes, by calculating $\Delta C_P^{\ddagger}$ values for parts of the structures (by recalculating energies and their variances for specific regions only; Fig. 3). Energy contributions from interactions with neighbouring regions are not included, and therefore one should not expect these 'partial' $\Delta C_P^{\ddagger}$ values to add-up to the total value. They do, however, offer new quantitative insights. Conceptually, one may expect differences in partial $\Delta C_P^{\ddagger}$ values to align with regions that differ in flexibility. This is

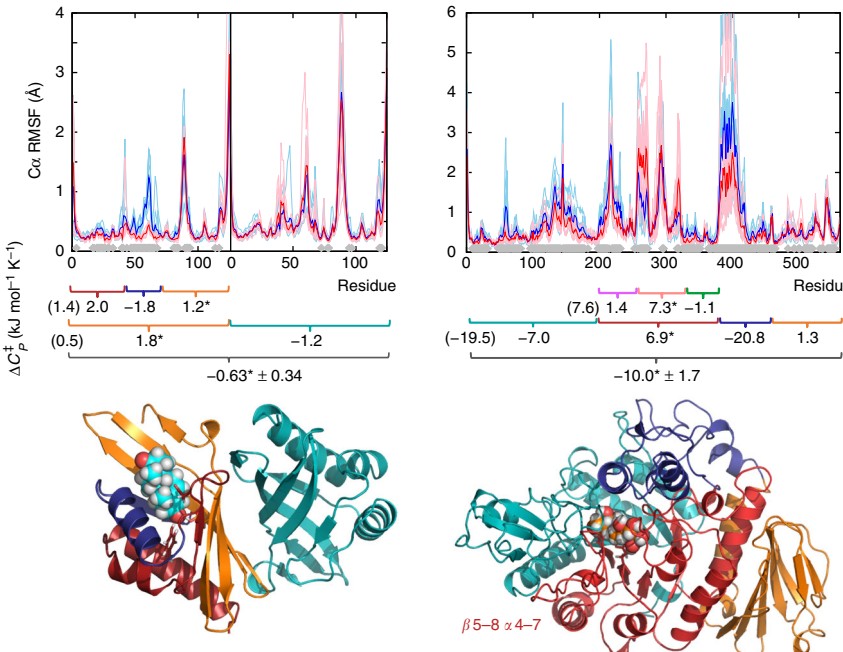

**Fig. 3** Structural fluctuations and partial heat capacity differences between reactant state and transition-state analogue complexes. Top: root-mean square fluctuations from 50–500 ns MD simulations for KSI (left) and MalL (right). Thin lines are individual runs, thick lines the average of ten runs. Residues for which the Cα RMSF difference between states are significant ($p < 0.01$ as determined by a two-sample $t$-test) are indicated by grey diamonds (full data in Supplementary Fig. 7). Middle: calculated partial $\Delta C_P^{\ddagger}$ values for protein regions. Values including contribution from the ligand are indicated (*). Bottom: illustration of KSI (left) and MalL (right) coloured based on partial $\Delta C_P^{\ddagger}$ regions from the middle pane. Standard deviations are indicated for the total $\Delta C_P^{\ddagger}$ values; see Supplementary Table 5 for residue ranges and standard deviations for partial $\Delta C_P^{\ddagger}$ values. Transition-state analogues are shown with space-filling spheres

largely true for some small regions with clear differences in flexibility (e.g. residues 46–70 for KSI; residues 250–321 and 374–459 for MalL; see Fig. 3), but is not obvious throughout the structure, especially for larger regions. Crucially, differences in $\Delta C_P^{\ddagger}$ are distributed across the full protein structure, whereas significant differences in flexibility are limited to regions that interact with the ligand bound in the active site. This observation bears similarity with the findings of Homans and others regarding entropy differences upon protein–ligand binding: unfavourable entropic contributions (restricted protein dynamics) around the binding site were observed to be (partially) offset by increases in the amplitude of motions in adjacent protein regions[15,16].

KSI, as a dimer, offers the opportunity to assess the dynamical role of the monomer that is distal to substrate turnover. The distal monomer of KSI is the main contributor to reduce $\Delta C_P^{\ddagger}$ at the TS (Fig. 3). Overall, the catalytic monomer contributes a positive $\Delta C_P^{\ddagger}$; the N- and C-terminal regions forming the back of the active site and more remote regions contribute a positive $\Delta C_P^{\ddagger}$, while helix 48–59 that closes over the active site opposite from the catalytic Asp38 rigidifies and contributes a negative $\Delta C_P^{\ddagger}$. The finding that the non-catalytic chain is as a significant contributor to negative $\Delta C_P^{\ddagger}$ points to an important role for the oligomer in the temperature dependence of the catalytic process. Enzyme oligomerization is common, indicating an evolutionary advantage[25]; however, the functional purpose of these quaternary interactions is not well understood. If interactions are optimised to allow global contributions from changes in the distribution of vibrational modes across the multimer[26], oligomerization may provide a means to tune the temperature dependence of rates through global contributions to the overall $C_P$ change.

The active site of MalL (and TIM barrel enzymes in general[27]) sits displaced to one side above the TIM barrel core, interacting

with a half of the barrel comprising β5–8 and α4–7 (Fig. 3). Analogous to the ligand bound chain of KSI, the catalytic half of the TIM barrel increases in $C_P$ at the TS, while more remote protein components, including the second TIM half barrel contribute to the overall negative $\Delta C_P^{\ddagger}$. The lid domain, consisting of a helix–loop–helix extension above the barrel, contributes significantly to the overall reduction in $C_P$ at the TS, consistent with a role in shielding the active site from solvent at the catalytic step. The parallels between the KSI dimer and MalL barrel halves are especially noteworthy in that the TIM barrel is argued to have evolutionary origins as a dimer of (βα)₄ units[13,27], the dynamical origins of which may still be discernible in the now fused structure.

Overall, these data indicate that the decrease of $C_P$ between the enzyme–substrate and enzyme–TS complexes is not just a function of rigidification of elements around the active site, but significant contributions are also made by regions remote from the active site, including oligomeric partners. Further, the individual contributions of different domains are markedly different in sign and magnitude (distinct from homogeneous rigidification), suggesting a functional and evolutionary role of spatially distant regions in thermoadaptation.

## Discussion

In enzyme catalysis, $\Delta C_P^{\ddagger}$ is emerging as a critical parameter for describing the temperature dependence of enzymatic rates, and as a consequence, for thermal adaptation in enzyme evolution[6,7]. The capacity to predictably manipulate enzyme activity with temperature continues to be a sought-after goal in biotechnology[28], but a lack of understanding of the principles governing thermal activity hampers the guided development of enzymes. The in silico replication of experimental $\Delta C_P^{\ddagger}$ values gives insight

into the atomic-level details of $C_P$ changes along the reaction coordinate, which govern the temperature dependence of enzyme rates. In turn, this provides a route to engineer temperature optima of enzymes: modifications in enzyme structure that change $\Delta C_P^{\ddagger}$ can be proposed and tested. Due to the difficulty of converging the difference in enthalpy variance that underlies $\Delta C_P^{\ddagger}$, one cannot expect perfect quantitative agreement between simulation and experiment, but trends and, importantly, atomistic mechanistic details can be extracted.

In two distinct enzyme systems, contributions to reduce $\Delta C_P^{\ddagger}$ at the TS come from small domains surrounding the active site, as well as domains distant from the catalytic centre. Rigidification of loops close to the active site is expected in the TS ensemble because of stabilisation of the TS. Unexpectedly, domains distal to the active site contribute significantly to the overall negative $\Delta C_P^{\ddagger}$, offsetting positive contributions to $\Delta C_P^{\ddagger}$ around the active site (excluding the loops). This observation has implications for the biological importance of both enzyme mass and oligomerization. Previously, enzyme mass has been found to be correlated to catalytic efficiency[6]. Enzyme mass is also directly related to its heat capacity, because adding amino acids to the protein increases the overall heat capacity[29]. Therefore, one could posit the conjecture that heat capacity is correlated to catalytic efficiency in some way. Note that we do not suggest that enzyme dynamics directly contributes to lowering of the energy barrier, or increasing the rate, of reaction (any such effects are likely to be small[30]); we do not investigate the reaction itself in this work. Here, simulation and experiment concur in showing negative values of $\Delta C_P^{\ddagger}$ for two enzymes, revealing (and identifying the nature of) significant differences in dynamical behaviour between the ground state and TS ensembles. Analysis of the contributions to these differences show that negative contributions to $\Delta C_P^{\ddagger}$ are dispersed throughout the protein and arise from auxiliary domains (MalL) and dimeric units (KSI) not directly involved with the reaction chemistry. The conjecture that the heat capacity of the enzyme is correlated with the catalytic efficiency and the observation that changes in heat capacity are dispersed across the enzyme is intriguing and suggests a range of further experiments; it may indicate a significant functional role of distal domains regardless of proximity to the active site, suggesting a reason for driving the evolution of these domains and interactions.

## Methods

**Enzyme production and characterisation.** Cloning, expression, purification and activity assays of MalL were as described previously[7]. Co-crystallisation of MalL with 0.5 mM 1-deoxynojirimycin was performed using hanging drop vapour diffusion at 18 °C. Crystals were flash cooled with cryoprotectant comprising of the crystallisation mixture with 20% glycerol for collection at the Australian Synchrotron (MX1). Molecular replacement was performed with the WT MalL apo structure (PDB 4M56 (https://doi.org/10.2210/pdb4M56/pdb))[7] as the search model.

The KSI sequence (*Pseudomonas testosteroni*) with a C-terminal hexa-His tag was optimised for expression in *Escherichia coli*. Expression was carried out over ~24 h in Luria-Bertani broth at 28 °C. Purified KSI was obtained by a two-step immobilised metal affinity chromatography-gel filtration chromatography process. KSI activity was measured in vitro using a continuous enzyme assay following the isomerization of 19-nor-androst-5(10)-ene-3,17-dione at 248 nm in phosphate buffer (pH 7.0) for minimal pH change with temperature. See Supplementary Note 1 for additional details for both enzymes.

**Experimental $\Delta C_P^{\ddagger}$ determination.** Temperature versus rate profiles were determined by measuring rates in a continuous assay at temperature intervals of 2–4 °C at saturating substrate concentrations. Temperature was controlled via a ThermoSpectronic single cell peltier, and independently checked before and after assays by thermocouple. Initial rates were measured over a period of 10 s to limit the effect of denaturation, if present, at elevated temperatures. Temperature profiles were fit with Eq. 3 with reference temperature ($T_0$) set to $T_{opt} - 4$ K:

$$\ln(k) = \ln\left(\frac{k_B T}{h}\right) - \frac{\left[\Delta H_{T_0}^{\ddagger} + \Delta C_P^{\ddagger}(T - T_0)\right]}{RT} + \frac{\left[\Delta S_{T_0}^{\ddagger} + \Delta C_P^{\ddagger}\ln(T/T_0)\right]}{R}, \quad (3)$$

where $k$ = rate; $k_B$ = Boltzmann constant; $h$ = Planck's constant; $\Delta H_{T_0}^{\ddagger}$ = enthalpy change at $T_0$; R = ideal gas constant; $\Delta S_{T_0}^{\ddagger}$ = entropy change at $T_0$. The transmission coefficient, $\kappa$, is assumed to be 1 for simplicity and is thus not included.

**MD simulation and analysis.** All simulations and analyses were performed using the Amber package (http://ambermd.org/) and the ff99SB-ILDN protein force field [31] (see also Supplementary Note 2). Tests with a more recent force field are included in Supplementary Note 3, Supplementary Fig. 3 and Supplementary Fig. 9. For KSI, PDB entry 1OHP (https://doi.org/10.2210/pdb1OHP/pdb) was used with Asn38 mutated back to the wild-type Asp and either the substrate or intermediate of the KSI reaction (Fig. 2a) modelled in chain A (based on the co-crystallised inhibitor 5α-estran-3,17-dione); chain B was left empty. General Amber FF (GAFF) parameters with charges from HF/6-31G(d) restrained electrostatic potential (RESP) fitting (RED server: http://upjv.q4md-forcefieldtools.org/REDS/) were used for the ligand. Asp38 was treated as protonated only for the intermediate state in chain A. Asp99 was protonated in both chains, with all other ionisable residues in their standard states. All three histidines were singly protonated on Nε2 and some Asn/His residue side-chains were rotated by 180° to obtain an optimal hydrogen bond network. For MalL, chain A from the 1-deoxynojirimycin bound structure obtained here (PDB entry 5WCZ (https://doi.org/10.2210/pdb5WCZ/pdb)) was used with missing atoms built in with COOT based on electron density where available. The substrate isomaltose was placed in the active site by overlay with PDB entry 3AXH (https://doi.org/10.2210/pdb3AXH/pdb) (E277A MalL from *Saccharomyces cerevisiae*; Cα RMSD 0.84 Å) and simulated in GLYCAM (06j-1) parameters. The TS analogue used in simulation was placed in the active site based on the modelled position of isomaltose and the position of 1-deoxynojirimycin in our co-crystal structure, and simulated using GLYCAM for the glucose unit, and GAFF (with HF/6-31G(d) RESP fitted charges) for the unit containing the protonated nitrogen. Asp63 and Glu371 were treated as protonated in both states, with the catalytic residues Asp199 unprotonated and Glu255 protonated (in line with the mechanism). Other ionisable residues were in their standard protonation states, with His161 singly protonated on Nδ1 and all other His on Nε2.

The preparation/equilibration protocol was as follows: solvation in a truncated octahedral box of TIP4P-Ew water molecules (keeping all crystallographic waters) with Na⁺ ions added to neutralise overall charge (ion positions randomised for each independent run), brief minimisation followed by heating in 20 ps to 300 K (KSI) or 320 K (MalL) with positional restraints on Cα atoms (5 kcal mol⁻¹ Å⁻²), gradual release of restraints in 40 ps, equilibration in the NPT ensemble for 1 ns. Five-hundred nanosecond production simulations were performed in the NVT ensemble with the Berendsen thermostat and loose temperature coupling (10 ps time constant). For both enzymes, restraints were used to maintain the Michaelis complex (with equivalent restraints on the IS or TSA states; see Supplementary Note 3).

Analysis was performed using 10 ps snapshots from 50–500 ns of the simulations, with force-field energies recalculated after stripping of solvent and ions. Clustering on the Cα RMSD (excluding the highly flexible C-terminal residues 117–125 in KSI and the N-terminal residues 1–6 in MalL) was performed as follows: for KSI, the *K*-means clustering algorithm was used to produce two clusters (after establishing the trajectories are best represented by two main conformational clusters, see Supplementary Table 3). For MalL, the hierarchical agglomerative algorithm was used with a minimum cluster distance of 2.1. Cα RMSF was calculated using RMSD fitting to a running average coordinates from a time window of 10 ns.

**Data availability.** Coordinates and structure factors for MalL co-crystallised with 1-deoxynojirimycin are deposited in the PDB under accession number 5WCZ. Simulation input files are available from figshare (https://doi.org/10.6084/m9.figshare.5875734).

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

## Acknowledgements

M.W.v.d.K. is a BBSRC David Phillips Fellow (BB/M026280/1) and he and A.J.M. thank the BrisSynBio Synthetic Biology Research Centre for funding (BB/L01386X/1). E.J.P. was supported by a University of Waikato Doctoral Scholarship and V.L.A. was supported by the Marsden Fund of New Zealand (16-UOW-027). M.C. and A.J.M. thank the EPSRC Centre for Doctoral Training in Theory and Modelling in Chemical Sciences (EP/L015722/1). We further acknowledge EPSRC funding for CCP-BioSim (EP/M022609/1). This work was conducted using the computational facilities of the Advanced Computing Research Centre, University of Bristol. We thank the reviewers for their insightful comments.

## Author contribution

M.W.v.d.K. devised simulation and analysis; M.W.v.d.K. and E.J.P. performed simulations and analysis, assisted by M.C.; E.J.P. and K.L.K. performed experiments; M.W.v.d. K., E.J.P., A.J.M. and V.L.A. analysed and interpreted results, and wrote the manuscript. M.W.v.d.K. and E.J.P. contributed equally to this work.

## Additional information

**Competing interests:** The authors declare no competing interests.

