## [Peer Review File(PDF 666 kb) · Nature Communications]

Reviewer #1 (Remarks to the Author):

This is a very nice, short but elegant paper from van der Kamp and coworkers, using a combination of atomistic molecular dynamics simulations and experiment to probe the dynamical origins of heat capacity changes in enzyme catalysed reactions, and to show how they can be predicted by simulations. This is valuable both for our understanding of fundamental biochemistry, and also from a methodological point of view. The authors are experts in their fields, and the work is well-executed and a pleasure to read, apart from some typographic and formatting issues, primarily in the SI, which need to be fixed through careful proofreading. There are some still issues I believe the authors should address, which I have enumerated below. However, if these can be satisfactorily addressed, I would be happy to recommend the manuscript for publications in Nature Communications.

1. Supporting Information, restraints in simulations: The restraints needed to keep D38 sound like a typical problem we have frequently observed when using older AMBER force fields such as ff99SB-ILDN, which the authors used in this paper (mainly due to the behaviour of flexible loops in the simulations with this force field – the authors note also that this may be a force field issue). Looking at the KSI crystal structure used by the authors, D38 is at the start of a potentially flexible loop – when the authors write that D38 swings out into solvent do they mean that the whole loop opens up or just the D38 side chain? In the case of the latter, the use of a more recent AMBER force field such as AMBER14SB should mitigate this problem, and it's worth at least testing.

2. I recommend the authors carefully proofread the main text and SI. For example, just in the section mentioned above, there are numerous typos, and formatting issues (deprotonated subscript, the . between kcal mol⁻¹ Å⁻² should be centered, quasi)harmonic etc – these should be cleaned up.

3. Pg. 7: “Examination of individual runs shows an opening near the MalL active site as loops surrounding the active site move apart. This is observed in simulations of both the substrate and TS analogue bound state. Movement in loops 213-221, 387-417, and 287-302 away from the active site creates an ‘opened’ structure associated with larger RMSD (Supplementary Figure 7c-e).” – this could possibly be the force field artefact mentioned in point 1 above. Can the authors please validate with a force field like AMBER14SB that this is not just because of the force field – loops have a tendency to be unphysically floppy in ff99SB-ILDN and open up a lot and just changing the force field tends to be enough to make the system more stable. At least validating this is quite important as linking changes in heat capacity to changes in dynamical properties is central to the paper – it is crucial that they are being described correctly by the force field on these long simulation time scales. (Note that unless changing the force field causes major differences I am not suggesting rerunning all simulations in the paper, just some spot validation so that one can be confident about the force field).

As a final note, the authors provide a commendable amount of simulation details, which is critical for reproducibility of the work.

Reviewer #2 (Remarks to the Author):

This is an impressive effort to use structure and simulation to reveal the origin of heat capacity in catalysis. The authors are able to reproduce experimental temperature dependence of catalytic rates. A notable claim is that remote regions of the protein may contribute to the energetics of catalysis. I am skeptical. Here are some things for the authors to be clearer about and address directly.

There remains a deep skepticism in the literature that there is a direct connection between the general fluctuations of the protein and the chemistry of catalysis (see Warshel's view: J. Chem. Phys. 144, 180901). Can the authors provide such a connection here? Or are they simply correlating the microscopic with the macroscopic?

The authors choose a convenient definition of the heat capacity that does not require an actual temperature dependence i.e. variance of the enthalpy at a given temperature. This is very unsatisfying, especially given a general inability of molecular dynamics to quantitatively reproduce experimental measurements of protein dynamics and the extreme sensitivity of energies of "hard" modes. There is very little direct microscopic contact with the macroscopic measurement of catalytic rate(T). The systems are relatively complicated, especially for KIS where two activated states are considered. Thus it seems prudent to examine the alternate definition of $C_p = TdS/dT$, which will provide direct contact with the experimental data, give a strong check of single temperature analysis, and help expose the origins of the heat capacity more fully. Without this additional information and analysis a critical reader will remain completely unconvinced.

The analysis of enthalpy is also somewhat unclear. In the supplementary material it is stated "Analysis was performed using 10 ps snapshots from 50-500 ns of the simulations, with force-334 field energies re-calculated after stripping of solvent and ions." Does this mean that protein-solvent interactions were ignored? If so, this is a serious flaw.

Furthermore, the illustrations and discussion focus on the alpha skeleton for structural analysis, which is very unsatisfying. It is not made clear what elements of the protein were used to calculate the heat capacity. All atoms? Or just the backbone? Experiment suggests that the side chains have

little T dependence and that most of the heat capacity is on the backbone (Nature Nature 411, 501; PNAS 114, 6563). Is this what the authors see? The general consensus in the protein literature is that the vast majority of the Cp resides in the backbone and protein-solvent interactions. More analysis is required. Hard versus soft (torsion) modes. Secondary structure contributions to Cp. etc.

The simulations required unusual restraints to prevent “drift” of the structures, particularly in the active site. The effects of this are not explained or the approach really justified. It would seem to be potentially quite a distortion. There are other simplifying technical approximations made that are justified by sweeping assertions only.

The G-H equation adapted to rate processes assumes a two state equilibrium. It is used without comment or justification. See below.

The authors make a point of noting that the simulations were done at temperatures removed from thermal unfolding implying that global unfolding is not relevant. This is likely true. However, it is well known that proteins have significant (i.e. large scale) subglobal and local motions (see QRB 40, 287), which could impact the temperature dependence of catalysis. Indeed, the loop reorganizations that are pointed to by the authors are examples of this.

There is also an ambiguity in definition – positive heat capacity changes are associated with the curvature seen in Figure 1 – as seen in any protein unfolding study and simulated in ref. 10, for example. Further, the authors assume a transmission coefficient of one!, which cannot be and is likely temperature dependent and invisible to their calculations.

In the abstract, it is stated “tightening of loops around the active site is observed as expected, but crucially, changes in energetic fluctuations are evident across the whole enzyme ... distal to the active site.” Not sure why the former is “expected” nor is the latter unusual as it has been seen dozens of times by NMR in response to ligand binding, including TS analogues.

Reviewer #3 (Remarks to the Author):

The authors perform extensive molecular dynamics simulations on models of enzyme-substrate and enzyme-intermediate (as a proxy for enzyme-TS) complexes, calculate molecular mechanics energy variances from these, use these as proxies for enthalpy variances, and from these use the standard statistical thermodynamics relationship to predict values for ΔCP of enzyme activation. Calculated values are in good agreement with experimental data, and the authors then go on in the discussion section to use the molecular models to provide suggestions at the atomic-level into the origins of mechanisms that might have evolved in proteins to modulate this thermodynamic parameter to their advantage.

The simulation methodology is state-of-the-art, and the simulation set-up and data analysis procedures are described in excellent detail, such that it would be possible to reproduce the experiments done.

There are a few areas that would benefit from some attention:

1. As has been discussed by many, including Prabhu and Sharp (reference 10), most analysis of heat capacity, and heat capacity changes, in protein systems divides the issue into two components: solvation terms, and what one could generalise as 'structural dynamic' terms. In the work described here, the enthalpy variances are equated to molecular mechanics energy variances, and though the simulations have been performed with full consideration of solvent (and ions), the molecular mechanics energies are actually re-evaluated a posteriori, without consideration of the contribution of either. The authors need to do a bit of work to convince their audience that this is a valid approach, and does not potentially neglect important terms. For example, one effect of solvent is to provide electrostatic screening. If the simulation involves significant fluctuations in the separation between two charged groups, then by ignoring the effect of the solvent on the dielectric constant, the resultant fluctuation in the electrostatic energy for this interaction will be over-estimated.
2. The authors do not hide the complexity of the data analysis process. One issue they have had to deal with is the dynamical instability of the simulations. Different replicates sample different regions of conformational space to differing degrees. They use an RMSD-based clustering approach to help deal with this. There is always an element of arbitrariness about clustering, and some form of sensitivity analysis is needed, and indeed maybe more justification for doing it at all. For example, if the cluster definition was made tighter a larger number of clusters, each containing structures of lower conformational variance, would be produced. It is likely that reduced conformational variance would correlate with reduced intra-cluster energetic variance, and so, since total variance is calculated as a population-weighted average of the cluster variances, this would decrease too. There is, obviously, no clustering going on in the experiment – how can it be justified here?
3. In the discussion the authors take advantage, as they should, of the atomistic detail the simulations provide to drill down into the data and look for patterns and trends. Though they do not mention it, the approach bears considerable similarity to the studies from the Homans group (and others) on the phenomenon of "entropy-entropy compensation" in protein ligand binding (see, e.g.

Bingham, JACS, 2004, 126, 1675-1681). This is particularly evident when one remembers that an alternative definition of C_p is the RMSF of the entropy, divided by k (see, e.g. eq 3 in reference 10). Thus, for example, figure 3 in this work may be compared with figures 2 and 3 in Roy Biophys J 2010, 99, 218-226 that studies the dynamics of the major urinary protein (MUP). Roy showed that, at the residue level, most apparent variations in protein flexibility on ligand binding, based on an MD simulation approach not dissimilar to the one used here, were not statistically significant. Now admittedly those simulations were done eight years ago and the present work features a greater number of longer replicate simulations, but it is still important that errors are quoted for the ΔC_p values shown in figure 3 in the present work so that the subsequent analysis can be shown to be meaningful.

4. The discussion includes a section on page 13 concerned with the concept of 'energy reservoirs'. I have two problems with this: firstly, at first reading the concept seems to go against the laws of thermodynamics since it appears to suggest the possibility of one part of a system at equilibrium having more (useable) energy than another, so it needs some more detailed explanation. Secondly, a clearer and fuller explanation is required as to how this hypothesis relates to the observations made in this work about domain-specific patterns in variation in ΔC_p – the connection is not obvious to me.

If these issues can be addressed, I think this will be a very significant piece of work.

Response to reviewer comments

Reviewer 1

The reviewer states that our manuscript is “a very nice, short but elegant paper” which is “valuable both for our understanding of fundamental biochemistry, and also from a methodological point of view”. The reviewer asserts that “the work is well-executed and a pleasure to read”.

Specific reviewer comments:

1. Supporting Information, restraints in simulations: The restraints needed to keep D38 sound like a typical problem we have frequently observed when using older AMBER force fields such as ff99SB-ILDN, which the authors used in this paper (mainly due to the behaviour of flexible loops in the simulations with this force field – the authors note also that this may be a force field issue). Looking at the KSI crystal structure used by the authors, D38 is at the start of a potentially flexible loop – when the authors write that D38 swings out into solvent do they mean that the whole loop opens up or just the D38 side chain? In the case of the latter, the use of a more recent AMBER force field such as AMBER14SB should mitigate this problem, and it’s worth at least testing.

The reviewer asks for some additional detail and raises the important point (which applies to any molecular dynamics simulation) of the choice and effects of the force-field. We have performed further simulations to test this. These tests show that our results are not significantly affected by the choice of force field.

As the referee points out, we described what was observed in the original SI when no restraints are applied (e.g. Supplementary Figure 3 legend). We include further details here. In the initial 1 ns pressure equilibration, the Asp38 side-chain moves out into solvent, followed by further movement of the loop away from its crystallographic position, not unexpected for a surface loop in solution. The latter consistently happens within the first ~10-50 ns of simulation. We have added this additional detail to the SI section “Restraints in simulations” as follows (new text in red):

“Without any restraints, the deprotonated Asp38 swings out into solvent **within the initial 1ns NPT equilibration ($dOD_{2_{Asp38}-C4_{substrate}}$ increased to 6.6-10.1; Supplementary Figure 3); once solvent exposed, the reactivity is severely limited¹³. Only after this initial change, instability of the loop that contains Asp38 occurs: the main-chain hydrogen bonds from Gly37 and Val40 (in the loop) to Ala114 are lost in the first 10 ns of simulation (donor-acceptor distances increase from <4 Å after NPT equilibration to >4.7 Å after 10 ns in all cases).**”

To test the possible dependence of the results on the force field, and to address whether the behaviour noted above can be mitigated by using a more recent force field, we have now further run test simulations of the reactant state (2 independent runs of 50 ns) without restraints with the Amber ff14SB force field. Essentially, the behaviour is the same as observed with ff99SB-ILDN and the observed instability is thus not just an artefact of the ff99SB-ILDN force field used.

We have added a description of these additional tests with the ff14SB force-field to the Supplementary Information. To aid this description, we have further added additional panels and text in the legend to Supplementary Figure 3.

Added text to SI in the section “*Restraints in simulations*”:

“**To address whether the observed behaviour can be mitigated by using a more recent, improved protein force field, simulations of the reactant state (2 independent runs of 50 ns) without restraints were performed with Amber ff14SB. Essentially, the behaviour is the same as observed before (with ff99SB-ILDN): initial movement of Asp38 away from the substrate into solvent happens within the initial 1ns NPT equilibration, and subsequently, the loop becomes less stable. Although the loop hydrogen bonds and structure may change less rapidly than with ff99SB-ILDN, hydrogen bonds still break within the first ~50 ns of simulation (Supplementary Figure 3d). This indicates that the movement of Asp38 into solvent followed by instability of the loop is not just an artefact of the ff99SB-ILDN force field used.**”

Additional text (in red) in Supplementary Figure 3 legend:

Supplementary Figure 3. Restraints required to maintain a Michaelis complex conformation in KSI simulations. **a**, Example snapshot from simulation of KSI with the reactant bound without restraints, indicating that Asp38 and the surrounding loop swing away from the bound substrate, into solvent. Distance between Asp38 C γ and C4 on the substrate is indicated, alongside the hydrogen bond donor-acceptor distances between Tyr14, Asp99 and O3 (see Supplementary Figure 1). Hydrogens are omitted for clarity. **b**, Starting structure for simulation (with inhibitor bound, from PDB 1OHP) with distances on which restraints are placed to keep Asp38 in a reactive conformation indicated with dashed lines. Distances measured in the structure are shown in black, and distance where the one-sided harmonic restraint comes into effect in red. **c**, Example snapshots (ff99SB-ILDN, without restraints) indicating the sequential movement of Asp38 moving out into solvent (orange backbone, with the loop still anchored by backbone hydrogen bonds between Glu37-Ala114 and Val40-Ala114 – taken after 1 ns NPT equilibration), prior to further loop movement (dark red, taken after 10 ns of production simulation). Starting structure in green. **d**, Example distance plots in simulations with ff99SB-ILDN, ff14SB and ff99SB-ILDN with restraints. Change in loop structure (C α RMSD of residues 37-42 after alignment on residues 1-116) and the presence/loss of backbone hydrogen bonds between Glu37-Ala114 and Val40-Ala114 (donor-acceptor distance) are indicated.

2. I recommend the authors carefully proofread the main text and SI. For example, just in the section mentioned above, there are numerous typos, and formatting issues (deprotonated subscript, the . between kcal mol⁻¹ Å⁻² should be centered, quasi)harmonic etc – these should be cleaned up.

We kindly thank the reviewer for pointing out some typographic and formatting issues (primarily in the SI) and we have now carefully proof-read and corrected them.

3. Pg. 7: “Examination of individual runs shows an opening near the Mall active site as loops surrounding the active site move apart. This is observed in simulations of both the substrate and TS analogue bound state. Movement in loops 213-221, 387-417, and 287-302 away from the active site creates an ‘opened’ structure associated with larger RMSD (Supplementary Figure 7c-e).” – this could possibly be the force field artefact mentioned in point 1 above. Can the authors please validate with a force field like AMBER14SB that this is not just because of the force field – loops have a tendency to be unphysically floppy in ff99SB-ILDN and open up a lot and just changing the force field tends to be enough to make the system more stable. At least validating this is quite important as linking changes in heat capacity to changes in dynamical properties is central to the paper – it is crucial that they are being described correctly by the force field on these long simulation time scales. (Note that unless changing the force field causes major differences I am not suggesting rerunning all simulations in the paper, just some spot validation so that one can be confident about the force field.)

We thank the reviewer for this comment regarding the Supplementary Information section “*Analysis of conformational sampling in Mall and outlier identification*”. We agree it is worthwhile to check if the loop movements observed are not simply a consequence of force-field choice. We have thus run additional simulations with the ff14SB force field (2 runs of 500 ns for the transition state analogue complex). Although C α RMSD from the starting structure tends to be lower with ff14SB, there is still significant flexibility in the loop regions indicated: C α RMSF indicates that this flexibility lies within the range sampled by the 10 runs with ff99SB-ILDN. We can thus conclude that that the loop movement and flexibility observed in ff99SB-ILDN simulations is not a force-field artefact. Because we are interested in the *difference* in heat capacity between the two states, we believe the ff99SB-ILDN simulations are valid; any effect of increased flexibility should equally affect the simulations of both states. We have added Supplementary Figure 8 with a comparison of C α RMSF from the two new ff14SB simulations with the original ff99SB-ILDN simulations.

Reviewer 2

The reviewer states that our work is “an impressive effort to use structure and simulation to reveal the origin of heat capacity in catalysis”. The reviewer is sceptical regarding “a notable claim [...] that remote regions of the protein may contribute to the energetics of catalysis”. We would like to make clear that we do not suggest or claim here that remote regions of the protein contribute to catalysis itself (i.e. lowering of the energy barrier). Our results indicate that remote regions are involved in determining the difference in heat capacity between different states, which is related to the *temperature dependence* of ΔH^\ddagger and ΔS^\ddagger (and the temperature optimum) of the enzymatic reaction rate. This conclusion is supported by the good agreement we find between simulation and experimental results. Below, we address the specific points that the reviewer makes.

1. There remains a deep skepticism in the literature that there is a direct connection between the general fluctuations of the protein and the chemistry of catalysis (see Warshel’s view: J. Chem. Phys. 144, 180901). Can the authors provide such a connection here? Or are they simply correlating the microscopic with the macroscopic?

As alluded to above, we do not claim a connection between fluctuations of the protein and the *chemistry* of catalysis: we only simulate defined molecular states that are representative of the species / transition states that are stabilised by the enzyme active sites in catalysis. We use molecular dynamics simulations to sample the conformational and energetic fluctuations of the enzymes in the different states, to calculate the difference in heat capacity, ΔC^\ddagger , which is a thermodynamic quantity. We claim no dynamical contribution to catalysis, and our simulations do not investigate that possibility (see also our theoretical discussion in Arcus *et al.* Biochemistry, 2016, 55, 1681–8. DOI: 10.1021/acs.biochem.5b01094). (NB We generally agree with the scepticism displayed by, amongst others, Prof Warshel, and would argue that the contribution of dynamics to *catalysis*, i.e. lowering of the energy barrier of reaction, is very small, if anything – and some of us have addressed this in previous work: Luk *et al.*, “Unraveling the role of protein dynamics in dihydrofolate reductase catalysis”, *Proc Natl Acad Sci USA*. 2013, 110, 16344-9. DOI: 10.1073/pnas.1312437110.) In our current work, we show agreement between the ΔC^\ddagger values calculated from the experimental values for the enzymatic reaction rate vs. temperature, we can thus correlate “the microscopic with the macroscopic”, which leads to new insights into the influence of enzyme structural fluctuations on enzyme temperature dependence.

2. The authors choose a convenient definition of the heat capacity that does not require an actual temperature dependence i.e. variance of the enthalpy at a given temperature. This is very unsatisfying, especially given a general inability of molecular dynamics to quantitatively reproduce experimental measurements of protein dynamics and the extreme sensitivity of energies of “hard” modes. There is very little direct microscopic contact with the macroscopic measurement of catalytic rate(T). The systems are relatively complicated, especially for KIS where two activated states are considered. Thus it seems prudent to examine the alternate definition of $C_p = TdS/dT$, which will provide direct contact with the experimental data, give a strong check of single temperature analysis, and help expose the origins of the heat capacity more fully. Without this additional information and analysis a critical reader will remain completely unconvinced.

The definition we use is indeed convenient but is also rigorous. It makes calculation of ΔC^\ddagger from simulations feasible. This approach has been applied previously in other contexts (see below), and is both theoretically justified and practically applicable. The reviewer’s comments do raise interesting points that go the heart of our paper and are pertinent, however. The reviewer states that there is “a general inability of molecular dynamics to quantitatively reproduce experimental measurements of protein dynamics”. This is not a fair reflection of the general view in this field; the ability of molecular dynamics simulations with recent force-fields (such as the one we have used) indicate a good correlation with a variety of experimental measurements (e.g. Perez *et al.*, *Curr Opin Struct Biol.*, 2016, 36, 25-31. doi: 10.1016/j.sbi.2015.12.002), although there is still room for improvement (see e.g. Maier *et al.*, *J Chem Theory Comput*, 2015, 11, 3696-713. doi: 10.1021/acs.jctc.5b00255; Koes & Vries, *Proteins* 2017 85, 1944-56. doi: 10.1002/prot.25350). Furthermore, we specifically

test the ability of molecular dynamics simulations to reproduce key experimental data in our work. The agreement between experiment and simulation here is remarkably good. In terms of using the alternative definition for $C_p = T(dS/dT)_p = (dU/dT)_p$, this is certainly another line of reasoning. Indeed, one of us has used this approach experimentally to show agreement between the temperature dependence of the enzyme catalysed rate and “transition state analogue” binding (see ref. 9: Firestone et al. 2017, *ACS Chemical Biology* **12**, 464-473). However, we argue that it is not currently feasible to examine this definition based on entropy using detailed atomistic molecular dynamics simulation. To expand on this point: To calculate heat capacity differences between (subtly) different protein states, measurement must be sensitive and significant sampling is required to get (close to) convergence for the calculated values. As we indicated in our paper, and is clear from previous work, this requires many μ s of simulation for either enthalpy or entropy estimations (see e.g. Baron *et al.*, 2009 *JCTC* **5**, p. 3150-3160). Thus, to use $C_p = T(dS/dT)_p$ would multiply the number of simulations needed from 20 for each enzyme (as in our manuscript) to at least 120 for each enzyme (20 at 6 different temperatures). In principle, one could calculate the heat capacity via the variance of the entropy without having to simulate at different temperatures, i.e. via $C_p = \frac{(\partial S^2)}{k_B}$ (and similarly, $\Delta C_p^\ddagger = \frac{\Delta(\partial S^2)^\ddagger}{k_B}$). In both cases, however, the difficulty is that typical protocols with which to calculate entropy from simulations often make significant approximations (e.g. assuming that order parameters are correctly represented by the force-field, and subsequently assuming that bond vector orientations are independent from other degrees of freedom and bond vector motions are axially symmetric; see Akke *et al.*, *J. Am. Chem. Soc.* 1993, **115**, 9832-9833), or need the use of additional, non-trivial simulation approaches (e.g. thermodynamic integration, Peter *et al.*, *J Chem Phys* 2004, **120**, 2652). Such approximations or additional simulations will have errors that may be larger than the subtle difference between the closely related protein-ligand states examined in this work. This is why in our work, and in previous works by experts in the field (e.g. Piana et al. 2012, *PNAS*, doi: 10.1073/pnas.1201811109; Fenley et al. 2012, *PNAS*, doi: 10.1073/pnas.1213180109), the variance of the enthalpy (or force-field energy) has been used to calculate heat capacity.

3. The analysis of enthalpy is also somewhat unclear. In the supplementary material it is stated “Analysis was performed using 10 ps snapshots from 50-500 ns of the simulations, with force-field energies re-calculated after stripping of solvent and ions.” Does this mean that protein-solvent interactions were ignored? If so, this is a serious flaw. Furthermore, the illustrations and discussion focus on the alpha skeleton for structural analysis, which is very unsatisfying. It is not made clear what elements of the protein were used to calculate the heat capacity. All atoms? Or just the backbone? Experiment suggests that the side chains have little T dependence and that most of the heat capacity is on the backbone (*Nature* **411**, 501; *PNAS* **114**, 6563). Is this what the authors see? The general consensus in the protein literature is that the vast majority of the C_p resides in the backbone and protein-solvent interactions. More analysis is required.

Protein-solvent interactions were treated properly during our explicit solvent molecular dynamics simulations. However, energy variances are indeed calculated in the absence of solvent (initially, see further below), due to the difficulty of including solvent accurately. We were thus relying on the assumption that the effect of protein-solvent interactions are similar for both states, and a sufficiently accurate *difference* in heat capacity, ΔC_p^\ddagger , can be calculated. We agree that, ideally, solvent effects should be taken into account, and we have therefore performed additional calculations and analysis and included this in our work (e.g. in the new Supplementary Table 5, see further below). We get quantitatively similar results for ΔC_p^\ddagger (when using an implicit solvent model) for both enzymes. We describe the significant additional work as well as issues with calculating protein-solvent effects in the response to Reviewer 3 (comment 1), who commented on our approach and protein-solvent interactions in detail. We have added the key points of this discussion as additional text to the Supplementary Information (see response to reviewer 3, comment 1).

We thank the reviewer for pointing out that experiment suggests that “most of the heat capacity is on the backbone”. Although for clarity, the illustrations primarily use (a cartoon representation of) the C α backbone, we write that “the force-field potential energy was used as an approximation for the system

enthalpy, and was recalculated for the protein-ligand system”, i.e. we use the full force-field potential energy function, including *all* protein and substrate atoms. We have clarified this, as well as the use of explicit solvent in the simulations, in the main text (section “*Heat capacity differences from simulation*”) as follows:

“A total of 5 μ s of **explicit solvent** MD simulation was run for KSI and MalL in both the substrate-bound and proxy TS representations over ten replicate simulations for each state (Supplementary Results). The force-field potential energy was used as an approximation for the system enthalpy, and was recalculated for the protein-ligand system (**i.e. all atoms**) without explicit water.”

Prompted by the comments from the reviewer, we have now also calculated the variance of the backbone only, i.e. the force-field potential energy function with all amino-acids stripped back to Glycine. Interestingly, the ΔC_p^\ddagger values calculated for the backbone only are similar but somewhat smaller than for the total system (-0.21 ± 0.04 and -7.6 ± 0.6 $\text{kJ}\cdot\text{mol}^{-1}\cdot\text{K}^{-1}$ for KSI and MalL, respectively), confirming that indeed a large part (but not all) of the heat capacity comes from the variance of the backbone energies. As well as adding this new data to Supplementary Table 5 (see also below in response to Reviewer 3 comments 1 and 3), we have added a note about these results in the main manuscript (section “*Local and global contributions to ΔC_p^\ddagger* ”) as follows:

“... allows meaningful analysis of the differences between the two ensembles. **A significant part of ΔC_p^\ddagger resides in the protein backbone (in agreement with experiments that suggest side-chains contribute only a fraction to the total protein heat capacity [Caro et al., Proc Natl Acad Sci 2017, 114, 6563-8]; Supplementary Table 5), although the contribution of side-chains cannot be ignored.**”

4. The simulations required unusual restraints to prevent “drift” of the structures, particularly in the active site. The effects of this are not explained or the approach really justified. It would seem to be potentially quite a distortion. There are other simplifying technical approximations made that are justified by sweeping assertions only.

We note that restraints are only used for KSI; the single restraint on the substrate MalL simulations only contributes to the force-field energy very occasionally, helping to maintain the substrate in the active site. In general, we aim to capture the ‘reactive’ or Michaelis complex, and therefore need to ensure that we do not sample diffusion processes related to formation of this complex. Justification of the approach used is included in the Supplementary Information, which we believe is appropriate. The restraints for KSI and their justification are now discussed in detail (see our response to reviewer 1, comment 1, above). We have thus endeavoured to refine our descriptions of the restraints to avoid “sweeping assertions” and have rewritten the text accordingly. Approximations regarding the cluster-weighted variance calculation are now justified in detail, see response to reviewer 3, comment 2.

5. The G-H equation adapted to rate processes assumes a two state equilibrium. It is used without comment or justification.

The theoretical background and justification of our approach is presented in detail in our paper Arcus *et al.* Biochemistry, 2016, 55, 1681-88. doi: 10.1021/acs.biochem.5b01094). Our approach is based on transition state theory, not Grote-Hynes (G-H) Theory. As noted above, we do not here aim to calculate reaction rates, and certainly not dynamical corrections to rates. The manuscript cites three equations all of which are standard from statistical thermodynamics and transition state theory, which does not require an assumption of a (pseudo) equilibrium between states – only an assumption of a dividing surface (see e.g. Truhlar, “Transition state theory for enzyme kinetics.” Arch Biochem Biophys, 2015, 582, 10-17. doi: 10.1016/j.abb.2015.05.004.).

6. The authors make a point of noting that the simulations were done at temperatures removed from thermal unfolding implying that global unfolding is not relevant. This is likely true. However, it is well known that proteins have significant (i.e. large scale) subglobal and local motions (see QRB 40, 287), which could impact the temperature dependence of catalysis. Indeed, the loop reorganizations that are pointed to by the authors are examples of this.

Our simulations and experiments indeed show no evidence of unfolding at these temperatures. The reviewer is quite correct that it is well known that proteins have significant subglobal and local motions. These local motions contribute to, and are captured by, the heat capacity term and it is precisely the influence of these local motions on the temperature dependence of the rate that we are addressing here. More specifically, we are capturing the changes in the distribution of local motions along the reaction coordinate.

7. There is also an ambiguity in definition – positive heat capacity changes are associated with the curvature seen in Figure 1 – as seen in any protein unfolding study and simulated in ref. 10, for example. Further, the authors assume a transmission coefficient of one!, which cannot be and is likely temperature dependent and invisible to their calculations.

The reviewer is not correct here regarding positive heat capacity changes. Figure 1 shows *negative* heat capacity changes are associated with curvature in the temperature dependence of the enzyme catalysed rate (see also theoretical discussion in Arcus *et al.* Biochemistry, 2016, 55, pp 1681–1688). The reviewer *is* correct that protein unfolding studies show a large and positive heat capacity change, quite distinct from the behaviour here. Those previous studies of folding (e.g. reference 10) are at equilibrium and consider the difference between the folded state and the unfolded state and say nothing about the transition state. In the case of CI2, the *rate* of protein unfolding shows a very small positive heat capacity change that is not significantly different from zero. This is due to the transition state for protein unfolding lying very close to the folded state on the reaction coordinate (see Tan *et al.*, 1996, *J Mol Biol.* **264**, 377-89). We do not discuss and do not seek to address folding here. The reviewer is correct that we assume that the transmission coefficient is one for the fitting of our experimental rate-temperature data (as stated in the Methods section). Whilst it is likely true that the transmission coefficient is not exactly equal to 1, this information is inaccessible to us either by experiment or from our simulations. Estimating contributions from recrossing, tunnelling and deviations from Boltzmann distributions (all contributors to the transmission coefficient) is not possible to achieve with accuracy in our systems. Furthermore, we would argue that we cannot simply add another variable to the equation (used for fitting our data to obtain the experimental ΔC_p^\ddagger) without this prior knowledge and so it is much safer to fix the transmission coefficient at 1. This is justified as the contributions from the transmission coefficient are thought to be small relative to other factors for enzymes (see e.g. Garcia-Viloca, Gao, Karplus & Truhlar, 2004, *Science.* **303**, 186-95). Indeed, Warshel and others have demonstrated that dynamical corrections to TST for enzyme-catalysed reactions are small (see also the discussion in Luk *et al.* 2013, Proc Natl Acad Sci 110, 16344).

8. In the abstract, it is stated “tightening of loops around the active site is observed as expected, but crucially, changes in energetic fluctuations are evident across the whole enzyme ... distal to the active site.” Not sure why the former is “expected” nor is the latter unusual as it has been seen dozens of times by NMR in response to ligand binding, including TS analogues.

We have rephrased the sentence in the abstract to remove subjective descriptors identified: “The simulations reveal subtle and surprising underlying dynamical changes: tightening of loops around the active site is observed ~~as expected, but crucially,~~ along with changes in energetic fluctuations ~~are evident~~ across the whole enzyme including important contributions from oligomeric neighbours and domains distal to the active site.”

We have further added a note regarding changes in response to ligand binding observed by NMR in the introduction:

“Further, the simulations provide an atomically detailed picture of the dynamical differences between transition state and Michaelis complexes that gives rise to this behaviour, revealing complex, and intriguing changes in dynamics across the whole enzyme structure (similar to changes observed by NMR in protein dynamics upon ligand binding [Bingham *et al.*, *J Am Chem Soc*, 2004, 126, 1675-81; MacRaild *et al.*, *J Mol Biol* 2007, 368, 822-32; Caro *et al.*, *Proc Natl Acad Sci* 2017, 114, 6563-8]).”

Reviewer 3

We are grateful to the reviewer for the generally supportive and insightful comments. The comments have helped to improve our work. The reviewer asserts that, if the issues (s)he points out can be addressed, “this will be a very significant piece of work”.

Below, we address the issues the reviewer raises in detail.

1. As has been discussed by many, including Prabhu and Sharp (reference 10), most analysis of heat capacity, and heat capacity changes, in protein systems divides the issue into two components: solvation terms, and what one could generalise as ‘structural dynamic’ terms. In the work described here, the enthalpy variances are equated to molecular mechanics energy variances, and though the simulations have been performed with full consideration of solvent (and ions), the molecular mechanics energies are actually re-evaluated a posteriori, without consideration of the contribution of either. The authors need to do a bit of work to convince their audience that this is a valid approach, and does not potentially neglect important terms. For example, one effect of solvent is to provide electrostatic screening. If the simulation involves significant fluctuations in the separation between two charged groups, then by ignoring the effect of the solvent on the dielectric constant, the resultant fluctuation in the electrostatic energy for this interaction will be over-estimated.

We thank the reviewer for the valid comments about the possible influence of solvent (especially for electrostatic screening). The question of solvent effects is a complex and interesting one. Here, we are interested in the *difference* in heat capacity, where an approach excluding solvent may well be appropriate – the potential overestimation of the fluctuation in the electrostatic energy would occur for both states, and would likely largely cancel out (due to the similarity of the conformational ensembles). Nevertheless, because of the possible importance of solvent effects, we originally attempted to calculate ΔC_p^\ddagger values based on energies obtained with a layer of explicit solvent (approximately the 1st and 2nd solvation shell) around the protein structures. However, there is no clear way to select which water molecules should be included in such a calculation. (The total potential energy of the full periodic system is not a good measure for *protein* heat capacity; we use an NVE-like NVT ensemble to limit the influence of thermo- and barostat on the dynamics.) An equal number of water molecules have to be included to meaningfully compare force-field energies, but this will lead to an ambiguous selection of water molecules and therefore differences in the distribution of water around the protein from one snapshot to the next. This will likely have the effect of overestimating the variance of the energies, in particular for the state that samples a larger conformational space (typically RS). Our initial results thus indicated an inflated ΔC_p^\ddagger . Apart from a ‘randomness’ of selecting which waters should be included, another issue of this approach is the potential errors that can be expected from standard water parameters for heat capacities and potential energies (see e.g. Levitt *et al.*, *J. Phys. Chem. B* 1997, 101, 5051-5061; with TIP4P, we can expect an error in heat capacities of at least 4 kJ·mol⁻¹·K⁻¹).

To avoid these issues with including explicit solvent in the energy calculation but still include the potentially important solvent effects such as electrostatic screening, we have now performed ΔC_p^\ddagger calculations based on force-field energies obtained with the Poisson-Boltzmann implicit solvent model. The results indicate similar ΔC_p^\ddagger values for both enzymes.

We refer to these results in the main manuscript as follows (new text in red):

“The force-field potential energy was used as an approximation for the system enthalpy, and was recalculated for the protein-ligand system without explicit water. Considering the *variance* of the enthalpy is the quantity required for C_p^\ddagger calculation (eq. (1)) and a *difference* in variance between two states is used to determine ΔC_p^\ddagger (eq. (2)) these approximations should be reasonable. **Calculating the variance with explicit solvent is problematic, because there is no clear criterion for selecting the water molecules that should be included in such a calculation (see Supplementary Information). We note that ΔC_p^\ddagger values calculated with an implicit solvent are qualitatively similar to those reported below for both enzymes (Supplementary Table 5).**”

As mentioned, the results themselves are included in Supplementary Table 5 (see further the response to comment 3). In addition, we have added the following discussion to Supplementary Information:

“*Calculation of variances and ΔC_p^\ddagger with solvent*”

The contribution of solvent to the variance of the force-field energies for the two states (from which ΔC_p^\ddagger is calculated) may be significant, e.g. due to electrostatic screening that solvent would provide. In an attempt to include possible solvent effects, we originally intended to calculate ΔC_p^\ddagger values based on variances of energies obtained with a layer of explicit solvent (approximately the 1st and 2nd solvation shell) around the protein structures. However, there is no clear way to select which water molecules should be included in such a calculation, and errors in the energies of water modelled by force-field approaches (see e.g. Levitt *et al.*, *J. Phys. Chem. B* 1997, 101, 5051-5061) may be a further source of error for the resulting heat capacities. We therefore used a Poisson-Boltzmann implicit solvent model (as implemented in Amber), see Supplementary Table 5. The resulting ΔC_p^\ddagger value for MalL converges less well with moving variance window size, but for both enzymes the values are qualitatively similar to those obtained without solvent. We can thus conclude that no large error is introduced by excluding contributions of solvent in calculation of the *difference* in variance between two states, from which ΔC_p^\ddagger is calculated.”

2. The authors do not hide the complexity of the data analysis process. One issue they have had to deal with is the dynamical instability of the simulations. Different replicates sample different regions of conformational space to differing degrees. They use an RMSD-based clustering approach to help deal with this. There is always an element of arbitrariness about clustering, and some form of sensitivity analysis is needed, and indeed maybe more justification for doing it at all. For example, if the cluster definition was made tighter a larger number of clusters, each containing structures of lower conformational variance, would be produced. It is likely that reduced conformational variance would correlate with reduced intra-cluster energetic variance, and so, since total variance is calculated as a population-weighted average of the cluster variances, this would decrease too. There is, obviously, no clustering going on in the experiment – how can it be justified here?

We thank the reviewer for pointing out potential issues with the use and specification of clustering for the analysis of variance for KSI (we do not use cluster-weighted variances for determining ΔC_p^\ddagger in MalL, as explained in the text).

Although no clustering is present in the experiment, we argue that, if substantially different conformational clusters (with substantially different variances) exist, it is relevant to analyse these clusters separately: the reaction itself (a proton transfer) will take place very rapidly (on the order of ~fs), so the enzyme conformation will not change from one cluster to another on this timescale. From experiment, we infer ΔC_p^\ddagger based on steady-state kinetics (and its dependence on temperature) from a very large number of individual enzyme molecules. If the simulations are sufficiently accurate, clustering will identify different relevant conformations (and their fractions) in the overall population of enzymes present in experiment.

The next question is what should constitute a cluster, or rather, in how many different clusters should the data be divided. We originally did a preliminary sensitivity analysis that led us conclude that for KSI, dividing into two clusters is the most relevant. We have now done this in more detail (leading to the same conclusion): we have performed hierarchical agglomerative clustering with different settings for the minimum distance between clusters, ϵ (clustering finishes when the minimum distance between clusters is greater than ϵ). The new Supplementary Table 3 indicates that only one or two main clusters are present for the two states (even though the total number of clusters is ≥ 10 for $\epsilon = 1.5$); any further minor clusters are at most 3.6% (typically much less) of the total.

We subsequently realised that, for a fair comparison, one should use the same clusters (and the same number of clusters) for each state and all conformations sampled should be included in these clusters: leaving out conformations that, using the hierarchical agglomerative algorithm, are clustered in minor clusters (e.g. 1% of data or below) could significantly affect the enthalpy variance, because by nature of the clustering, the outlier conformations (which may present outlier enthalpies) will be put in a different cluster.

We have thus used K-means clustering on the KSI simulation sets, in order to produce *exactly* two clusters, incorporating all snapshots/conformations sampled. These clusters are essentially the same as the main two clusters obtained with the hierarchical agglomerative algorithm: the centroids are

identical, and all conformations from the two main hierarchical agglomerative clusters are again separated out in the two K-means clusters.

The clustering details have now been modified in the Methods section to reflect this as follows:

“Clustering on the C α RMSD (excluding the highly flexible C-terminal residues 117-125 in KSI and the N-terminal residues 1-6 in MalL) was performed as follows: for KSI, the K-means clustering algorithm was used to produce two clusters (after establishing the trajectories are best represented by two main conformational clusters, see Supplementary Results). For MalL, the hierarchical agglomerative algorithm was used with a minimum cluster distance of 2.1.”

After re-calculating the ΔC_p^\ddagger using the cluster-weighted variances, the value changes slightly: -0.63 instead of -0.67 kJ·mol $^{-1}$ ·K $^{-1}$ for the total system, and similar small changes for the partial ΔC_p^\ddagger values (as reflected in updated versions of Figures 2 and 3, see also below).

In the SI, we have replaced the original section “*Combined conformational clustering of MalL simulations*” with an expanded section on “*Conformational clustering*” incorporating the discussion and results from above (new text in red):

“*Conformational clustering*”

For KSI, all conformations sampled were divided into two separate conformational clusters for both simulated states. This was done on the basis that for both states, only two highly similar significant clusters are present (Figure 2 and Supplementary Table 3). To ensure all sampled conformations are included in the analysis, the K-means algorithm was used (see Methods). Because in this case, substantially different conformational clusters exist (with substantially different variances; Figure 2), it is relevant to analyse these clusters separately: reactions take place very rapidly if the energy barrier to reaction is overcome (e.g. proton transfer on the order of \sim fs) and a transition state is (by definition) very short-lived. The enzyme conformation will thus not change from one cluster to another on this timescale. From experiment, ΔC_p^\ddagger is inferred from steady-state kinetics and its dependence on temperature, involving a very large number of individual enzyme molecules. If the simulations are sufficiently accurate, clustering will identify different relevant conformations (and their fractions) in the overall population of enzymes present in experiment.

Combined clustering of the 20 MalL simulations (10 for each state) using the hierarchical agglomerative algorithm with a minimum cluster distance (ϵ) of 2.1 Å leads to 9 clusters with an overall contribution $>1\%$ (Supplementary Figure 8). Only the top two clusters are sampled significantly in both states (of the remaining clusters, one has a 0.58% contribution of the minor contributing state; all others are 0.08% or less).”

We have added references to this new section at the appropriate locations in the main text where the cluster-weighted average of variance for KSI states is discussed.

3. In the discussion the authors take advantage, as they should, of the atomistic detail the simulations provide to drill down into the data and look for patterns and trends. Though they do not mention it, the approach bears considerable similarity to the studies from the Homans group (and others) on the phenomenon of “entropy-entropy compensation” in protein ligand binding (see, e.g. Bingham, JACS, 2004, 126, 1675-1681). This is particularly evident when one remembers that an alternative definition of C_p is the RMSF of the entropy, divided by k (see, e.g. eq 3 in reference 10). Thus, for example, figure 3 in this work may be compared with figures 2 and 3 in Roy Biophys J 2010, 99, 218-226 that studies the dynamics of the major urinary protein (MUP). Roy showed that, at the residue level, most apparent variations in protein flexibility on ligand binding, based on an MD simulation approach not dissimilar to the one used here, were not statistically significant. Now admittedly those simulations were done eight years ago and the present work features a greater number of longer replicate simulations, but it is still important that errors are quoted for the ΔC_p values shown in figure 3 in the present work so that the subsequent analysis can be shown to be meaningful.

We thank the reviewer for the suggestion of comparison to studies from the Homans group and others. We agree that there are important similarities. We have added a note and citations to key studies

regarding this in the introduction (see response to Reviewer 2, comment 8) and we have added a note on this to the appropriate paragraph in the main manuscript:

“Crucially, differences in ΔC_p^\ddagger are distributed across the full protein structure, whereas significant differences in flexibility are limited to regions that interact with the ligand bound in the active site. This observation bears similarity with the findings of Homans and others regarding entropy differences upon protein-ligand binding: unfavourable entropic contributions (restricted protein dynamics) around the binding site were observed to be (partially) offset by increases in the amplitude of motions in adjacent protein regions [Bingham *et al.*, *J Am Chem Soc* 126, 1675-1681 (2004); MacRaild *et al.*, *J Mol Biol* 368, 822-832 (2007)].”

We further thank the reviewer for the suggestion to compare to the work by Roy *et al.* In our case, with 10 independent 500 ns simulations per state, there are distinct regions where differences in flexibility (as indicated by $C\alpha$ RMSF) are indeed statistically significant. We have indicated this in a revised Figure 3 by a gray diamond for each residue where $p < 0.01$ for the difference in RMSF (as determined by a two-sample t-test between the two sets of $C\alpha$ RMSF measurements).

We have further added an additional supplementary figure (Supplementary Figure 7) showing the difference in $C\alpha$ RMSF between the two states (for each enzyme) and the p-value for that difference (similar to Figure 3 in Roy *Biophys J* 2010, 99, 218-226).

New Supplementary Figure 7 legend:

“**Supplementary Figure 7. $C\alpha$ RMSF differences between states and their significance.** **a**, $C\alpha$ RMSF difference for KSI RS and IS states. **b**, $C\alpha$ RMSF difference for MalL RS and TSA states. P-values at each residue are determined by a two-sample t-test between the two sets of $C\alpha$ RMSF measurements and plotted below. P-values < 0.01 are indicated in the top plot with gray diamonds.”

We agree with the reviewer that, in addition to the error bars shown in Figure 2f, it is important that errors are quoted for *all* ΔC_p^\ddagger values calculated. We have added errors to Figure 3 for the overall ΔC_p^\ddagger values, but to avoid Figure 3 becoming overly busy, we have added a table with all calculated (partial) ΔC_p^\ddagger values and their errors to the Supplementary Table 5 (see below); this has the additional benefit of clarifying the exact residue ranges used for the partial ΔC_p^\ddagger decomposition, as well as including the values obtained when calculating energies with implicit solvent (reviewer 2, comment 3, and reviewer 3, comment 1) and for the protein backbone only (reviewer 2, comment 4).

The Figure 3 legend has been updated to reflect these changes and refer to Supplementary Table 5:

“**Fig. 3. Structural fluctuations and partial variances between reactant state and transition state analogue complexes.** Top: root-mean square fluctuations from 50-500ns MD simulations for KSI (left) and MalL (right). Thin lines are individual runs, thick lines the average of 10 runs. Residues for which a RMSF difference between states is significant ($p < 0.01$ as determined by a two-sample t-test) are indicated by gray diamonds. Middle: calculated partial ΔC_p^\ddagger values for protein regions. Values including contribution from the ligand are indicated (*). Bottom: illustration of KSI (left) and MalL (right) colored based on partial ΔC_p^\ddagger regions from top pane. Standard deviations are indicated for the total ΔC_p^\ddagger values; see Supplementary Table 5 for residue ranges and standard deviations for partial ΔC_p^\ddagger values. Transition-state analogues shown in space-filling spheres.”

Supplementary Table 5. Calculated ΔC_p^\ddagger values ($\text{kJ}\cdot\text{mol}^{-1}\cdot\text{K}^{-1}$) and their standard deviations for different subsections of the two enzymes.

Ketosteroid isomerase			MalL		
Region	ΔC_p^\ddagger	Std. Dev. ^a	Region	ΔC_p^\ddagger	Std. Dev. ^b
Total (A+B)*	-0.63	0.34	Total (1-561*)	-9.97	1.72
Backbone ^c	-0.21	0.04	Backbone ^c	-7.57	0.61
Monomer A*	1.77	1.65	1-193	-7.02	0.88
Monomer B	-1.20	0.59	194-373*	6.92	1.84
1-45 (A)	1.96	2.62	374-459	-20.76	5.11

46-70 (A)	-1.80	2.04	460-561	1.31	0.13
71-125* (A)	1.15	1.23	194-249	1.36	0.30
			250-321*	7.29	2.50
			322-373	-1.08	0.11
Total* + PBSA ^c	-2.26	0.63	Total* + PBSA ^c	-3.84 ^d	0.19

* indicates that the ligand is included. ^aStandard deviation calculated as the sum of the standard deviations of RS and IS variances computed using a leave-one-out procedure (one simulation of each state is omitted at a time).

^bStandard deviation calculated from the sum of the standard deviations of RS and TSA moving variances for each of the 10 simulations. ^cVariances obtained from force-field energies calculated after deleting all amino-acid side-chains. ^dFor consistency, the moving variance window was again 70 ns (in line with the calculations without implicit solvent), but ΔC_p^\ddagger does not converge well with window size (with 60 or 80 ns windows, ΔC_p^\ddagger is -3.54 and -4.06 kJ·mol⁻¹·K⁻¹, respectively). ^ePBSA energy calculations were performed with sander from AmberTools 16. For computational efficiency, the ‘mbondi’ set of atomic radii was used, the total electrostatic energy was calculated with the particle-particle particle-mesh (P3M) procedure [164], and the cut-off distance for van der Waals interactions was 8 Å.

- The discussion includes a section on page 13 concerned with the concept of ‘energy reservoirs’. I have two problems with this: firstly, at first reading the concept seems to go against the laws of thermodynamics since it appears to suggest the possibility of one part of a system at equilibrium having more (useable) energy than another, so it needs some more detailed explanation. Secondly, a clearer and fuller explanation is required as to how this hypothesis relates to the observations made in this work about domain-specific patterns in variation in ΔC_p – the connection is not obvious to me.

We agree with the reviewer that the phrase “energy reservoirs” is problematic. We do not suggest that one part of the system has more (usable) energy than another (which we agree would go against the laws of thermodynamics). We have thus removed the phrase “energy reservoirs” and have rewritten this paragraph to be clearer about the conjecture that is being suggested, and the link with the observations we report. Note that we do not want to dive into the dynamics-catalysis debate, which is not appropriate based on the current work (see also responses to reviewer 2); we simply want to suggest that measuring ΔC_p^\ddagger for enzymes provides a possible route to investigate aspects of this debate.

The connection with the domain-specific patterns in variation of ΔC_p^\ddagger is the observation that differences in variance (and therefore ΔC_p^\ddagger) are distributed over the full protein structure, and thus reflect that the relevant differences in vibrational modes of the enzyme can contain contributions from the full protein structure, including distal domains or monomers. To add a fuller explanation of this kind, we have added to and adapted the relevant text in the manuscript as follows:

“Previously, enzyme mass has been found to be correlated to catalytic efficiency⁶. Enzyme mass is also directly related to its heat capacity as adding amino acids to the protein increases the overall heat capacity [Makhatadze, *Biophys Chem* 71, 133-156 (1998)]. Therefore, one could posit the conjecture that heat capacity is correlated to catalytic efficiency in some way. Here, through simulation and experiment we observe negative values of ΔC_p^\ddagger for two states on the reaction coordinate and this implies changes in the distribution of vibrational modes along the coordinate. Analysis of the contributions to these changes show that negative contributions to ΔC_p^\ddagger are dispersed throughout the protein and arise from auxiliary domains (MaIL) and dimeric units (KSI) not directly involved with the reaction chemistry. The conjecture that the heat capacity of the enzyme is correlated with the catalytic efficiency and the observation that changes in heat capacity are dispersed across the enzyme is intriguing and suggests a range of further experiments; it may indicate a significant functional role of distal domains regardless of proximity to the active site, suggesting a reason for driving the evolution of these domains and interactions.”

Reviewer #1 (Remarks to the Author)

This is a very nice, short but elegant paper from van der Kamp and coworkers, using a combination of atomistic molecular dynamics simulations and experiment to probe the dynamical origins of heat capacity changes in enzyme catalysed reactions, and to show how they can be predicted by simulations. This is valuable both for our understanding of fundamental biochemistry, and also from a methodological point of view. The authors are experts in their fields, and the work is well-executed and a pleasure to read, apart from some typographic and formatting issues, primarily in the SI, which need to be fixed through careful proofreading. There are some still issues I believe the authors should address, which I have enumerated below. However, if these can be satisfactorily addressed, I would be happy to recommend the manuscript for publications in Nature Communications.

1. Supporting Information, restraints in simulations: The restraints needed to keep D38 sound like a typical problem we have frequently observed when using older AMBER force fields such as ff99SB-ILDN, which the authors used in this paper (mainly due to the behaviour of flexible loops in the simulations with this force field – the authors note also that this may be a force field issue). Looking at the KSI crystal structure used by the authors, D38 is at the start of a potentially flexible loop – when the authors write that D38 swings out into solvent do they mean that the whole loop opens up or just the D38 side chain? In the case of the latter, the use of a more recent AMBER force field such as AMBER14SB should mitigate this problem, and it's worth at least testing.

2. I recommend the authors carefully proofread the main text and SI. For example, just in the section mentioned above, there are numerous typos, and formatting issues (deprotonated subscript, the . between kcal mol⁻¹ Å⁻² should be centered, quasi)harmonic etc – these should be cleaned up.

3. Pg. 7: "Examination of individual runs shows an opening near the MalL active site as loops surrounding the active site move apart. This is observed in simulations of both the substrate and TS analogue bound state. Movement in loops 213-221, 387-417, and 287-302 away from the active site creates an 'opened' structure associated with larger RMSD (Supplementary Figure 7c-e)." – this could possibly be the force field artefact mentioned in point 1 above. Can the authors please validate with a force field like AMBER14SB that this is not just because of the force field – loops have a tendency to be unphysically floppy in ff99SB-ILDN and open up a lot and just changing the force field tends to be enough to make the system more stable. At least validating this is quite important as linking changes in heat capacity to changes in dynamical properties is central to the paper – it is crucial that they are being described correctly by the force field on these long simulation time scales. (Note that unless changing the force field causes major differences I am not suggesting rerunning all simulations in the paper, just some spot validation so that one can be confident about the force field).

As a final note, the authors provide a commendable amount of simulation details, which is critical for reproducibility of the work.

Reviewer #2 (Remarks to the Author)

This is an impressive effort to use structure and simulation to reveal the origin of heat capacity in catalysis. The authors are able to reproduce experimental temperature dependence of catalytic rates. A notable claim is that remote regions of the protein may contribute to the energetics of catalysis. I am skeptical. Here are some things for the authors to be clearer about and address directly.

There remains a deep skepticism in the literature that there is a direct connection between the general fluctuations of the protein and the chemistry of catalysis (see Warshel's view: *J. Chem. Phys.* 144, 180901). Can the authors provide such a connection here? Or are they simply correlating the microscopic with the macroscopic?

The authors choose a convenient definition of the heat capacity that does not require an actual temperature dependence i.e. variance of the enthalpy at a given temperature. This is very unsatisfying, especially given a general inability of molecular dynamics to quantitatively reproduce experimental measurements of protein dynamics and the extreme sensitivity of energies of "hard" modes. There is very little direct microscopic contact with the macroscopic measurement of catalytic rate(T). The systems are relatively complicated, especially for KIS where two activated states are considered. Thus it seems prudent to examine the alternate definition of $C_p = TdS/dT$, which will provide direct contact with the experimental data, give a strong check of single temperature analysis, and help expose the origins of the heat capacity more fully. Without this additional information and analysis a critical reader will remain completely unconvinced.

The analysis of enthalpy is also somewhat unclear. In the supplementary material it is stated "Analysis was performed using 10 ps snapshots from 50-500 ns of the simulations, with force-334 field energies re-calculated after stripping of solvent and ions." Does this mean that protein-solvent interactions were ignored? If so, this is a serious flaw.

Furthermore, the illustrations and discussion focus on the alpha skeleton for structural analysis, which is very unsatisfying. It is not made clear what elements of the protein were used to calculate the heat capacity. All atoms? Or just the backbone? Experiment suggests that the side chains have little T dependence and that most of the heat capacity is on the backbone (*Nature* 411, 501; *PNAS* 114, 6563). Is this what the authors see? The general consensus in the protein literature is that

the vast majority of the Cp resides in the backbone and protein-solvent interactions. More analysis is required. Hard versus soft (torsion) modes. Secondary structure contributions to Cp. etc.

The simulations required unusual restraints to prevent “drift” of the structures, particularly in the active site. The effects of this are not explained or the approach really justified. It would seem to be potentially quite a distortion. There are other simplifying technical approximations made that are justified by sweeping assertions only.

The G-H equation adapted to rate processes assumes a two state equilibrium. It is used without comment or justification. See below.

The authors make a point of noting that the simulations were done at temperatures removed from thermal unfolding implying that global unfolding is not relevant. This is likely true. However, it is well known that proteins have significant (i.e. large scale) subglobal and local motions (see QRB 40, 287), which could impact the temperature dependence of catalysis. Indeed, the loop reorganizations that are pointed to by the authors are examples of this.

There is also an ambiguity in definition – positive heat capacity changes are associated with the curvature seen in Figure 1 – as seen in any protein unfolding study and simulated in ref. 10, for example. Further, the authors assume a transmission coefficient of one!, which cannot be and is likely temperature dependent and invisible to their calculations.

In the abstract, it is stated “tightening of loops around the active site is observed as expected, but crucially, changes in energetic fluctuations are evident across the whole enzyme ... distal to the active site.” Not sure why the former is “expected” nor is the latter unusual as it has been seen dozens of times by NMR in response to ligand binding, including TS analogues.

Reviewer #3 (Remarks to the Author)

The authors perform extensive molecular dynamics simulations on models of enzyme-substrate and enzyme-intermediate (as a proxy for enzyme-TS) complexes, calculate molecular mechanics energy variances from these, use these as proxies for enthalpy variances, and from these use the standard statistical thermodynamics relationship to predict values for ΔC_p of enzyme activation. Calculated values are in good agreement with experimental data, and the authors then go on in the

discussion section to use the molecular models to provide suggestions at the atomic-level into the origins of mechanisms that might have evolved in proteins to modulate this thermodynamic parameter to their advantage.

The simulation methodology is state-of-the-art, and the simulation set-up and data analysis procedures are described in excellent detail, such that it would be possible to reproduce the experiments done.

There are a few areas that would benefit from some attention:

1. As has been discussed by many, including Prabhu and Sharp (reference 10), most analysis of heat capacity, and heat capacity changes, in protein systems divides the issue into two components: solvation terms, and what one could generalise as 'structural dynamic' terms. In the work described here, the enthalpy variances are equated to molecular mechanics energy variances, and though the simulations have been performed with full consideration of solvent (and ions), the molecular mechanics energies are actually re-evaluated a posteriori, without consideration of the contribution of either. The authors need to do a bit of work to convince their audience that this is a valid approach, and does not potentially neglect important terms. For example, one effect of solvent is to provide electrostatic screening. If the simulation involves significant fluctuations in the separation between two charged groups, then by ignoring the effect of the solvent on the dielectric constant, the resultant fluctuation in the electrostatic energy for this interaction will be over-estimated.
2. The authors do not hide the complexity of the data analysis process. One issue they have had to deal with is the dynamical instability of the simulations. Different replicates sample different regions of conformational space to differing degrees. They use an RMSD-based clustering approach to help deal with this. There is always an element of arbitrariness about clustering, and some form of sensitivity analysis is needed, and indeed maybe more justification for doing it at all. For example, if the cluster definition was made tighter a larger number of clusters, each containing structures of lower conformational variance, would be produced. It is likely that reduced conformational variance would correlate with reduced intra-cluster energetic variance, and so, since total variance is calculated as a population-weighted average of the cluster variances, this would decrease too. There is, obviously, no clustering going on in the experiment – how can it be justified here?
3. In the discussion the authors take advantage, as they should, of the atomistic detail the simulations provide to drill down into the data and look for patterns and trends. Though they do not mention it, the approach bears considerable similarity to the studies from the Homans group (and others) on the phenomenon of 'entropy-entropy compensation' in protein ligand binding (see, e.g. Bingham, JACS, 2004, 126, 1675-1681). This is particularly evident when one remembers that an alternative definition of C_p is the RMSF of the entropy, divided by k (see, e.g. eq 3 in reference 10). Thus, for example, figure 3 in this work may be compared with figures 2 and 3 in Roy Biophys J 2010, 99, 218-226 that studies the dynamics of the major urinary protein (MUP). Roy showed that, at the residue level, most apparent variations in protein flexibility on ligand binding, based on an MD

simulation approach not dissimilar to the one used here, were not statistically significant. Now admittedly those simulations were done eight years ago and the present work features a greater number of longer replicate simulations, but it is still important that errors are quoted for the ΔC_p values shown in figure 3 in the present work so that the subsequent analysis can be shown to be meaningful.

4. The discussion includes a section on page 13 concerned with the concept of 'energy reservoirs'. I have two problems with this: firstly, at first reading the concept seems to go against the laws of thermodynamics since it appears to suggest the possibility of one part of a system at equilibrium having more (useable) energy than another, so it needs some more detailed explanation. Secondly, a clearer and fuller explanation is required as to how this hypothesis relates to the observations made in this work about domain-specific patterns in variation in ΔC_p – the connection is not obvious to me.

If these issues can be addressed, I think this will be a very significant piece of work.

Response to Reviewer 2

(reviewer's comments in blue; new/adapted text in red)

We thank Reviewer 2 for their further detailed reading of the manuscript, and their constructive criticism. The reviewer raises a number of interesting questions that are well worthy of discussion, although space limits what is possible in the manuscript. We have further modified our manuscript as described below.

The most important is how the calculations of heat capacity are actually done. It is clear, and acknowledged in this manuscript, that most of the C_p comes from the backbone and interactions with solvent. As acknowledged, the interaction with solvent is not well treated and leads to uncertainty in the veracity of the conclusions.

As noted, we performed additional simulations to investigate solvent effects and added discussion of solvent effects and contribution to the manuscript, and to the Supporting Information. Qualitatively, our conclusions are thus strengthened, while demonstrating that solvent effects are relevant, and also we added discussion of this important issue raised by this reviewer.

This is why I also raised the issue of the particular definition of C_p that was employed. While, in principle, the convenient definition used is theoretically rigorous, the simulation is not as it suffers from approximations and the general inability of current potentials to reproduce quantitatively experimental dynamics (this is absolutely clear despite what the authors assert - R^2 between experimental fast dynamics by NMR and simulation rarely and unpredictably reach > 0.7). Thus, using another definition allows a direct test or estimate of this reliability. That was my point. If the authors insist that using temperature variance to check the calculations is too hard then the ms needs to acknowledge the issue and the difficulty in doing the obvious calculation.

The definition we employ is indeed rigorous, as noted. The demonstrated agreement between our simulation results and experiments provides evidence in support of our approach. No simulation employing empirical potentials is entirely without limitations, of course, even when one accepts the limitations of classical molecular dynamics. The reviewer's central point is the possible dependence of the results on the empirical potential function employed. This is an important point, and we have tested this dependence in the revised manuscript, as described in our response and the revised Supporting Information (and noted by Reviewers 1 and 3). We have now also indicated this in the main manuscript (Methods section):

“All simulations and analyses were performed using the Amber package and the ff99SB-ILDN protein force field. Tests with a more recent force field are included in the Supporting Information.”

We agree that it would be interesting, in principle, to investigate other approaches for the calculations of heat capacities, but this is not feasible for enzymes with current computational resources. We would certainly test the reviewer's suggestion if we could! Unfortunately, this is not feasible (to be precise, reliable convergence cannot be achieved) even with significant high performance computer resources; the simulations we have performed for the current work have taken many months to complete.

The approach suggested by the referee (using either $C_p = T(dS/dT)_p$ over multiple temperatures or $C_p = \frac{(\partial S^2)}{k_B}$ at a single temperature) is unfortunately beyond what can currently be practically calculated/converged, as explained in our original response. It is well known that calculations of entropies and entropy changes from simulation converge particularly slowly. To address the reviewer's comments, we take their advice and have added two sentences to acknowledge this: one simply to state the point and a second to suggest some optimism that this might be achievable in the future.

“We note that ΔC_p^\ddagger values calculated with an implicit solvent are qualitatively similar to those reported below for both enzymes (Supplementary Table 5). Note that there is, in principle, an alternative approach to calculating ΔC_p^\ddagger , via the variance in entropy [10]. Calculating entropy from simulations accurately is much more challenging, however, although this may be feasible in the future.”

We acknowledge the important point made by the reviewer that classical simulations with current empirical potentials do not reproduce in detail experimental results for fast dynamics from NMR; this is widely known and discussed and certainly does not imply that useful results on protein dynamics cannot be extracted from simulations; we realise that the reviewer is not suggesting this. Also, it is important to note that we are of course here concerned with relative, not absolute, measure of protein dynamics, and thus cancellation of errors can be expected. We have now added a sentence to address the reviewer's comments in the manuscript:

"Such sampling is possible in molecular dynamics simulations with a 'molecular mechanics' description of the atoms and their interactions. Molecular mechanics force fields have been developed and optimized over many years,[Karplus, M. & McCammon, J. A. Nature Structural Biology 9, 646, doi:10.1038/nsb0902-646 (2002).] and can provide a generally good description of the structure and dynamics of proteins and protein-ligand binding.[Perez, A., Morrone, J. A., Simmerling, C. & Dill, K. A. Current Opinion in Structural Biology 36, 25-31, doi:https://doi.org/10.1016/j.sbi.2015.12.002 (2016).]. They are, however, empirical potential functions and typically (for reasons of computational efficiency) lack physically important effects such as variations in electronic polarization. This may be related to limitations in their ability to capture details of fast dynamics [O'Brien, E. S., Wand, A. J. & Sharp, K. A. Protein Science 25, 1156-1160, doi:10.1002/pro.2922 (2016).ref1]. Here, to calculate

ΔC_p^\ddagger , we use extensive molecular dynamics simulation to quantify the change in fluctuation between two states, A and B (Fig. 1a). The difference in heat capacity between these states can be determined by: [eq 2]"

It also seems odd to me to claim that the C_p measured by the temperature dependence of catalysis is unrelated to catalysis, which is effectively what is argued in the rebuttal. I think we are on the same side of the argument. It is just that the issue needs to be acknowledged. As long as the authors simply recognize this in the text then I have no objections.

We agree with the reviewer that we fundamentally agree on these issues, and we are happy to acknowledge this in the manuscript text. We are careful to avoid making unsubstantiated claims on the origins of catalysis or effects on catalysis. Our simulations do not compare catalysed and uncatalysed reactions, so do not address the origins of catalysis. The key point we intended to make in the response is that our current simulation study does not investigate the chemical reaction itself, and therefore cannot lead to a conclusion that enzyme dynamics contributes directly to increasing the rate of reaction. Accordingly, we have added a sentence and clarified the discussion to recognize this:

"Previously, enzyme mass has been found to be correlated to catalytic efficiency⁶. Enzyme mass is also directly related to its heat capacity; adding amino acids to the protein increases the overall heat capacity.²⁴ Therefore, one could posit the conjecture that heat capacity is correlated to catalytic efficiency in some way. **Note that we do not suggest that enzyme dynamics directly contributes to lowering of the energy barrier, or increasing the rate, of reaction (any such effects are likely to be small [Luk, L. Y. P. et al. Proc. Nat. Acad. Sci. 110, 16344-16349, doi:10.1073/pnas.1312437110 (2013)]); we do not investigate the reaction itself in this work. Here, simulation and experiment concur in showing negative values of ΔC_p^\ddagger for two enzymes, revealing (and identifying the nature of) significant differences in dynamical behavior between the ground state and transition state ensembles. Analysis of the contributions to these differences show [...]**"

Finally, the G-H equation doesn't require anything fancy to derive and understand. Positive heat capacity gives rise to a dG maximum and predicts cold (low T) and thermal (T) protein denaturation. In other words, concave down profiles, precisely what is seen in Fig. 1D. It is not "quite distinct" from what is discussed in this ms. For a particularly elegant presentation I refer the authors a giant of protein thermodynamics and statistical mechanics: Biopolymers 26, 1859. As I suggested previously, there is a definition issue and a lack of acknowledgement of assumptions. The authors need to be really clear on this and not simply point to ref. 6, which does an inadequate job on this point. With respect to definition, the curves will be the same for opposite sign of C_p because for protein unfolding the represented species is the folded state whereas for the rate process the relevant ref state is the activated state, which is the minor species. Second, the underlying assumption of the G-H equation as written is that C_p is temperature independent i.e. constant. What do the authors think about that? Seems important. T-dependence would reveal this

as well. The fact that the experimental curves conform to that assumption is probably all that is needed to be said though having simulations reproduce this would be better. Hardening the use of the G-H equation and making it clearer to the reader will make this ms better.

We agree with all that is stated by the reviewer here. By way of explanation, the distinction that we were referring to in our reply (our phrase “quite distinct”) is exactly that as described by the reviewer (in *italics and underlined* above) and nothing more. In our previously published work (Arcus *et al.* Biochemistry 2016), we stated: “The significance of ΔC_p for the temperature dependence of *protein stability* at equilibrium has been understood for 40 years.(33, 34) The role of ΔC_p^\ddagger in the temperature dependence of protein folding kinetics has also been well described.(35) For protein folding, ΔC_p^\ddagger is generally negative as the unfolded state is very dynamic and hydrophobic side chains have a characteristic solvation shell in this state, whereas the transition state for protein folding is generally relatively compact and more closely resembles the native state.(36) For folding, desolvation of hydrophobic residues on the folding pathway is thought to make a significant contribution to ΔC_p^\ddagger .(37) In contrast, the similarity between the folded state and the transition state for folding leads to small positive values of ΔC_p^\ddagger for protein unfolding and almost no curvature in the temperature dependence of unfolding rates.(35)” We did not, and do not, claim to have derived a novel thermodynamic relationship and agree with the reviewer’s entirely correct assertion regarding heat capacity changes and protein (un)folding. To address the reviewer’s first point we have added a sentence to the introduction:

“We have demonstrated that this accounts for the curvature observed in Eyring plots for a number of enzymes⁶. As we have discussed previously, curvature in Eyring plots due to ΔC_p^\ddagger has also been observed for protein folding kinetics [e.g. Tan, Y. J., *et al.* (1996) *J Mol Biol.* **264**, 377–389]. This is directly analogous to temperature-dependent curvature in protein stability due to ΔC_p^\ddagger that gives rise to both high- and low-temperature denaturation [Becktel, W. J. & Schellman, J. A. (1987). **26**, 1859-1877].”

Secondly, the referee raises a very interesting point regarding the possible T-dependence of ΔC_p^\ddagger , which we have indeed considered. The reviewer’s suggestion to make this explicit is useful. In this work, we assume that ΔC_p^\ddagger is temperature independent (at least for the range of temperatures studied). This assumption is based on, and justified by, the good fit of the curve to the experimental data. We have now made this assumption explicit in the text with an additional sentence and remain open to the idea of a temperature-dependent ΔC_p^\ddagger should the experimental data justify this; the results here indicate that the temperature dependence of ΔC_p^\ddagger is small or zero, justifying the approach here. We have modified the text as follows:

“Mall temperature dependence means that at lower temperatures, the rate approaches zero much faster for Mall than for KSI. Implicit in this approach is the assumption that ΔC_p^\ddagger is independent of temperature in the temperature range studied, and is justified based on the good fit of the MMRT model to the experimental data.”

Reviewer #1 (Remarks to the Author)

The authors have performed the additional simulations I requested, and provided thoughtful responses to my comments. The manuscript has been appropriately updated to reflect the issues I asked the authors to take into consideration. I am therefore satisfied with the revisions to the manuscript.

Reviewer #2 (Remarks to the Author)

Let me begin by stating that I found the original manuscript to be thought provoking and elegant. My comments were meant to prod clarification. For the most part the authors have done this though in some cases they continue to evade the issue somewhat. I will focus on those issues that I raised that the authors do not address and merely dismiss. The most important is how the calculations of heat capacity are actually done. It is clear, and acknowledged in this manuscript, that most of the C_p comes from the backbone and interactions with solvent. As acknowledged, the interaction with solvent is not well treated and leads to uncertainty in the veracity of the conclusions. This is why I also raised the issue of the particular definition of C_p that was employed. While, in principle, the convenient definition used is theoretically rigorous, the simulation is not as it suffers from approximations and the general inability of current potentials to reproduce quantitatively experimental dynamics (this is absolutely clear despite what the authors assert - R2 between experimental fast dynamics by NMR and simulation rarely and unpredictably reach > 0.7). Thus, using another definition allows a direct test or estimate of this reliability. That was my point. If the authors insist that using temperature variance to check the calculations is too hard then the ms needs to acknowledge the issue and the difficulty in doing the obvious calculation. It also seems odd to me to claim that the C_p measured by the temperature dependence of catalysis is unrelated to catalysis, which is effectively what is argued in the rebuttal. I think we are on the same side of the argument. It is just that the issue needs to be acknowledged. As long as the authors simply recognize this in the text then I have no objections. Finally, the G-H equation doesn't require anything fancy to derive and understand. Positive heat capacity gives rise to a dG maximum and predicts cold (low T) and thermal (T) protein denaturation. In other words, concave down profiles, precisely what is seen in Fig. 1D. It is not "quite distinct" from what is discussed in this ms. For a particularly elegant presentation I refer the authors a giant of protein thermodynamics and statistical mechanics: Biopolymers 26, 1859. As I suggested previously, there is a definition issue and a lack of acknowledgement of assumptions. The authors need to be really clear on this and not simply point to ref. 6, which does an inadequate job on this point. With respect to definition, the curves will be the same for opposite sign of C_p because for protein unfolding the represented species is the folded state whereas for the rate process the relevant ref state is the activated state, which is the minor species. Second, the underlying assumption of the G-H equation as written is that C_p is temperature independent i.e. constant. What do the authors think about that? Seems important. T-dependence would reveal this as well. The fact that the experimental curves conform to that assumption is probably all that is needed to be said though having simulations reproduce this would be better.

Hardening the use of the G-H equation and making it clearer to the reader will make this ms better. Congratulations on a fine effort!

Reviewer #3 (Remarks to the Author)

I am pleased to see that the authors have taken all my comments on board, and have undertaken a considerable amount of extra work to address issues I flagged up. As far as I can see, the results they get from the enhanced analysis continue to support their hypotheses, provide more robust statistical evidence for their significance, and some of the less defensible claims in the discussion have been toned down or rephrased in a satisfactory way.